# Proviral NUP153 binding to viral proteins and RNA regulates structural–nonstructural protein ratios in orthoflavivirus infection

Marie B. A. Peters [1,2], Richard Lindqvist[1,2], Eszter Kassa[3], Wai-Lok Yau[2,4], Pallabi Sengupta [5], Isabell Niedermoser [6], Gisa Gerold [6], Nasim Sabouri [5,7], Ylva Ivarsson [3], Richard Lundmark [2,4] & Anna K. Överby [1,2] ✉

Orthoflaviviruses are RNA viruses that cause serious diseases in humans, with currently no antivirals available. Targeting host factors is emerging as an attractive antiviral approach. However, as a first step, there is a need to understand which host proteins are hijacked and for what purpose. Here, using a combination of fluorescence microscopy, knock-down, crosslinking immunoprecipitation sequencing, mass spectrometry, and in vitro and biophysical assays, we identify nucleoporin-153 (NUP153) as a proviral factor during orthoflavivirus infection. We show that NUP153 is recruited to the virus amplification site on the endoplasmic reticulum to impact the structural to nonstructural viral protein ratios. We find that NUP153 interacts with both the viral proteins NS3 and NS5, and a highly conserved G-rich motif on the viral RNA. These interactions specifically promote the production of viral structural proteins, leading to an efficient virion assembly, virus release and spread to new cells. We propose that NUP153 acts as a key regulator in viral protein ratios, a mechanism that appears conserved among orthoflaviviruses.

Orthoflaviviruses, such as West Nile virus (WNV), yellow fever virus (YFV), dengue virus (DENV), tick-borne encephalitis virus (TBEV), and Zika virus (ZIKV)[1–3] are arthropod-borne RNA viruses that are responsible for high morbidity and mortality across the world. Although vaccines have been developed against some of these viruses, no clinically approved antivirals are currently available against them. Orthoflaviviruses are 40–60 nm spherical enveloped viruses that contain a single-stranded positive-sense RNA genome of approximately 11 kb, which is translated into a single polyprotein that is proteolytically processed into three structural proteins [the capsid (C) protein, the precursor form of the membrane protein (prM), and the envelope (E) protein] and seven nonstructural proteins (NS1, NS2A, NS2B, NS3, NS4A, NS4B, and NS5)[2,4,5]. Following entry into the cell,

orthoflaviviruses reprogram the host cellular machinery and induce major structural rearrangements of the endoplasmic reticulum (ER) to support viral replication, protein translation, and virion assembly, and we define this as the viral amplification site[4,6,7]. To accomplish the reprogramming, viral proteins hijack host proteins. Gaining insights into the mechanisms by which orthoflaviviruses exploit their host environment to promote viral propagation could lead to the identification of targets for host-directed therapies to combat infections.

Nucleoporins (NUPs) are the building blocks of the nuclear pore complex (NPC), which is one of the largest complexes in the cell, reaching a size of 120 MDa in human[8]. The NPC is composed of multiple copies of approximately 30 different NUPs[9]. The main function of the NPC is to create a channel across the nuclear envelope by fusing

[1]Department of Clinical Microbiology, Umeå University, Umeå, Sweden. [2]Laboratory for Molecular Infection Medicine Sweden (MIMS), Umeå University, Umeå, Sweden. [3]Department of Chemistry for Life Sciences, Uppsala University, Uppsala, Sweden. [4]Department of Medical and Translational Biology, SciLifeLab, Umeå University, Umeå, Sweden. [5]Department of Medical Biochemistry and Biophysics, Umeå University, Umeå, Sweden. [6]Institute of Virology, Medical University of Innsbruck, Innsbruck, Austria. [7]Science for Life Laboratory, Umeå University, Umeå, Sweden. ✉e-mail: anna.overby@umu.se

the inner and outer nuclear membranes for transport of proteins and RNA molecules across the nuclear membranes[10]. One of the NUPs, NUP153, is located at the nuclear basket of the NPC. It is involved in multiple RNA and protein export pathways, and is required for correct assembly of the NPC[11,12]. Interestingly, during hepatitis C virus (HCV) infection NUP153 is recruited to the ER-derived cytoplasmic structures where it regulates the transport of viral proteins into the HCV replication complexes[13]. Orthoflaviviruses also utilize ER membranes for their replication, and might also utilize NUP153 as a host factor during infection.

Here we demonstrate that NUP153 is a host factor exploited by both tick- and mosquito-borne orthoflaviviruses. Using a combination of fluorescence microscopy, knock-down, crosslinking immunoprecipitation sequencing, mass spectrometry, in vitro biophysical assays and mass spectrometry, our data shows that during flavivirus infection, NUP153 expression is upregulated and the protein is recruited to the viral amplification site on the ER, where it interacts with viral proteins NS3 and NS5, and a conserved G-rich motif on the viral RNA. These interactions impact the ratio of structural to nonstructural proteins to promote effective virion production, which is different from NUP153's proposed role during HCV infection[13]. Importantly, the results reveal a pro viral role of NUP153 in orthoflavivirus infection.

## Results

### Viral infection upregulates NUP153 expression and causes NUP153 relocation to the viral amplification site

NUP153 plays a role in the life cycle of some viruses[13,14], but its function during orthoflavivirus infection is poorly understood. To shed light on NUP153's role during orthoflavivirus infection, we first analyzed its localization during tick-borne orthoflavivirus infection, as the protein is typically found in the nuclear envelope (NE). We infected cells with TBEV and Langat virus (LGTV; commonly used as a BSL2 model for TBEV) as representative tick-borne orthoflaviviruses and analyzed the cellular localization of NUP153 using microscopy (Fig. 1A)[15]. We found that in the infected cells, the levels of NUP153 appeared increased and that a fraction of NUP153 colocalized with the viral NS3 protein in the cytoplasm. Quantification of the NUP153 signal revealed a 3-fold increase of NUP153 levels during TBEV and LGTV infection, both in the nucleus and the cytoplasm, as compared to the levels in the mock-treated samples (Fig. 1B, C). However, no difference in protein ratio between cytoplasm/nucleus was detected between mock and infected cells (Supplementary Fig. 1A). The upregulation of NUP153 at the level of RNA was verified in HEK 293T cells infected with both LGTV and TBEV (Supplementary Fig. 1B). Upregulation could be caused by either the virus or the host innate responses. To distinguish between these scenarios, we analyzed the RNA transcript of virus, interferon β (IFNβ), and NUP153 in A549 cells or A549 cells knocked out for the mitochondrial antiviral-signaling protein (MAVS), which is an adapter protein of importance for the innate immune defense against viruses[16]. TBEV replicated to the same extent in both cell types (Fig. 1H), and A549 cells readily upregulated IFNβ to high levels but not in cells with knocked out MAVS expression, in line with its expression being related to the innate immune response (Fig. 1I). NUP153 expression was instead significantly upregulated independently of MAVS expression (Fig. 1J). Furthermore, we confirmed that the conventional role of NUPs in the export of cellular mRNA was not compromised in infected cells (Supplementary Fig. 1C–F).

During flavivirus infection, the viral protein NS3 is present at the cytosolic side of the host ER, along with viral dsRNA, in the replication vesicles[6,17]. To determine if NUP153 is recruited to the ER during infection, we conducted super-resolution microscopy using structured illumination microscopy (SIM) on TBEV-infected cells (Fig. 1D, E). Interestingly, the fluorescent signal from both calnexin (ER marker) and NUP153 was increased in infected cells (Fig. 1D, E). The SIM analysis revealed that NUP153 protein has a reticular pattern in the cell which overlapped both with calnexin and viral NS3 during infection (Fig. 1D). We also found that a fraction of the NUP153 staining overlapped with dsRNA, the viral RNA replication intermediate, as a specific marker of RNA replication vesicles (Fig. 1E). Next, we quantified the amount of colocalization of NUP153, calnexin, NS3, and dsRNA (Fig. 1F, G). NS3 and NUP153 almost completely overlap with each other, and we found that both proteins also colocalized with dsRNA (Fig. 1G). Furthermore, our colocalization analysis also revealed that while most of the dsRNA signal colocalized with both NS3 and NUP153, most of NUP153 and NS3 did not colocalize with dsRNA, indicating that these proteins can also be found outside of the replication vesicles in the remaining of the ER as shown by their colocalization with ER marker calnexin (Fig. 1F). Collectively, the above data indicated that viral infection promotes ER expansion and ER localization of NUP153.

### NUP153 is a proviral factor during early infection

NUP153 is an essential cellular protein. To investigate its role in TBEV infection, we used the auxin-induced degron (AID) system to rapidly degrade endogenous neonGreen-tagged NUP153 in DLD-1 cells [NUP153(NG)^AID] without affecting the integrity of the NPC[18]. Using this system, DLD-1 cells were genetically modified to express AID and neonGreen-tagged NUP153, which can be rapidly degraded conditionally upon addition of auxin[18]. These cells were compared with DLD-1 control cells, which is the parental cell line. First, we evaluated the degradation efficiency, and full degradation of the NUP153(NG)^AID protein occurred within 2 h of auxin treatment (Fig. 2A, B, and Supplementary Fig. 1A, B). Secondly, we evaluated the cellular distribution using immunofluorescence analysis, and the neonGreen signal overlapped almost completely with the NUP153 signal in these cells, supporting that the neonGreen signal is a reliable marker for NUP153 localization (Fig. 2C, D). Further, we verified the recruitment and colocalization of NUP153 with NS3 at the viral amplification site in NUP153(NG)^AID and DLD-1 cells (Fig. 2E, F, and Supplementary Fig. 1C).

Using the AID system, we depleted NUP153 and analyzed the effect on LGTV infection. In these experiments, the infection level was quantified by measuring the amount of viral E-positive cells with immunofluorescence analysis and viral titers in the supernatant. We found that the viral infection was reduced by 50% in the absence of NUP153 (Fig. 2G) and viral titers were reduced by 80% (Fig. 2H). Furthermore, we analyzed the effect on TBEV infection after depletion of NUP153 at different timepoints to investigate the importance of NUP153 activity during the viral life cycle. We observed that the addition of auxin 2 h prior to infection, and at 2 hpi or 4 hpi resulted in approximately 40% reduction of infection in NUP153(NG)^AID cells but not in uninduced cells or the parental DLD-1 control cells (Fig. 2I and Supplementary Fig. 2D) and viral titers were reduced by 80−95% (Fig. 2J). Auxin addition at 6 hpi led to no significant reduction in the number of infected NUP153(NG)^AID cells, and had no detectable effect on the number of subsequently released viral particles (Fig. 2K and Supplementary Fig. 2E). Because we detected differences between the reduction in number of infected cells compared to the released viral particles following depletion of NUP153 we measured the amount of TBEV RNA at 2, 8, 12, and 16 h post infection in cells induced with auxin 1 hpi (Fig. 2L). We found that depletion of NUP153 led to approximately 50% reduction in TBEV viral RNA at 16 hpi, however, no reduction at 8 or 12 hpi (Fig. 2L), indicating that NUP153 depletion does not affect early replication but more likely affects virus assembly and release.

To verify that the reduction of viral infection was dependent on NUP153 expression, NUP153(NG)^AID cells were either transfected with plasmids expressing either enhanced green fluorescent protein (eGFP) or eGFP-tagged NUP153. Complete restoration of infected cells was seen by eGFP-NUP153 plasmid transfection (Fig. 2M). Taken together, these results indicated that although NUP153 is an important proviral factor during the early stages of viral infection, it does not affect viral

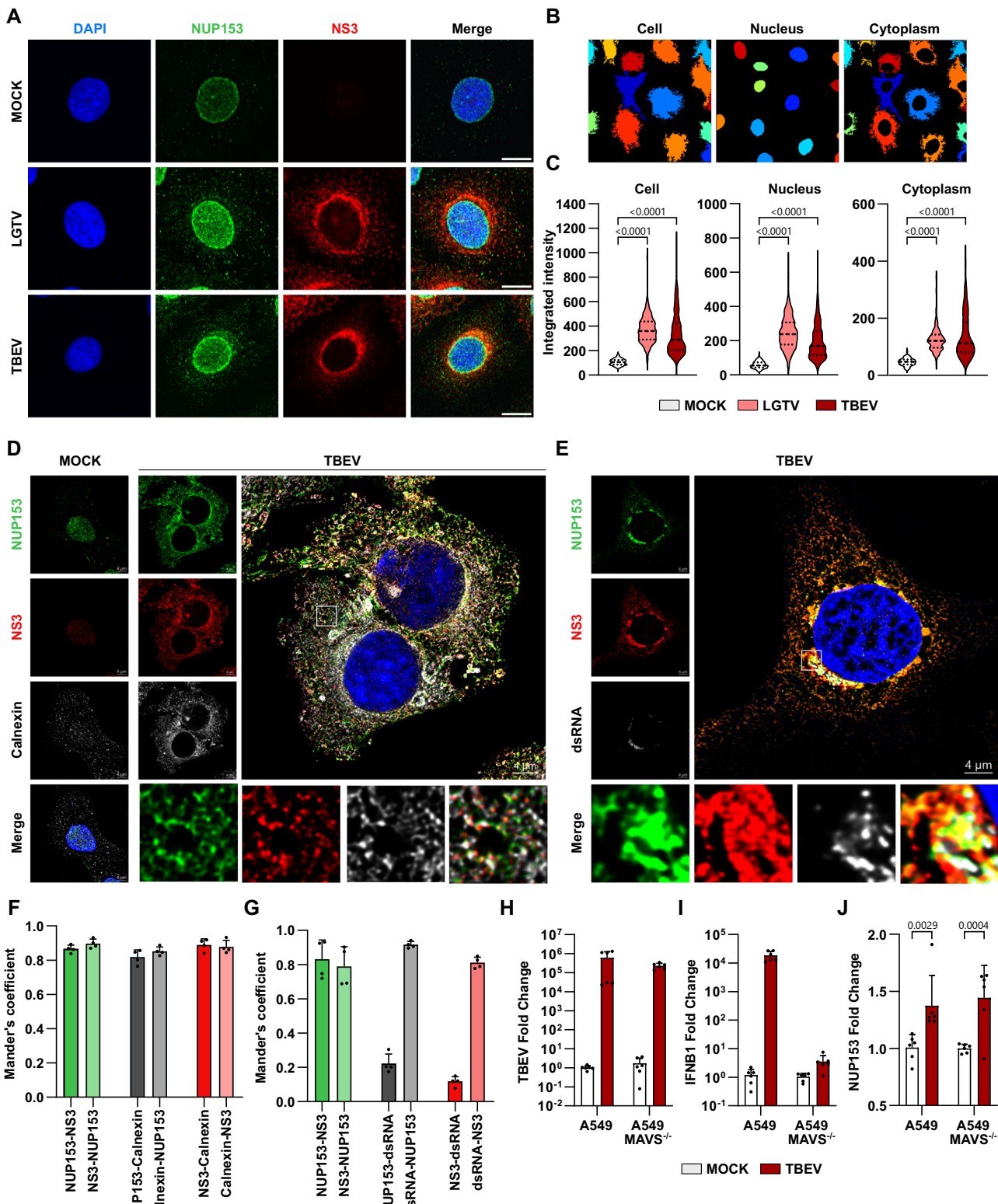

**Fig. 1 | TBEV infection alters NUP153 expression and cellular distribution.**
**A**–**C** Immunofluorescence analysis of A549 cells mock-treated or infected with TBEV or LGTV for 24 h. Representative confocal images of cells stained for DAPI (blue), NS3 (red), NUP153 (green) (**A**), Scale bars, 10 μm. **B** Example of CellProfiler-based segmentation into whole-cell, nucleus, and cytoplasmic compartments. **C** Quantification of NUP153 integrated fluorescence intensity in indicated compartments. Violin plots show individual cells (n = 303 [mock], 3, n = 310 [LGTV], and n = 368 [TBEV]). Statistical significance was determined by one-way ANOVA with Kruskal–Wallis test (P values shown in figure). **D**–**G** Structural illumination

microscopy (SIM) of A549 cells infected with TBEV for 24 h. Representative images stained for DAPI (blue), NUP153 (green), NS3 (red) and Calnexin (**D**) or dsRNA (**E**) (grays), scale bars, 4 μm. Colocalization analysis of independently acquired images using Mander's coefficient of **D** (**F**) and **E** (**G**) (n = 4 representing 4 acquired images). Data are presented as mean + s.d. RT-qPCR analysis of TBEV (**H**), IFNB1 (**I**) and NUP153 (**J**) RNA levels in A549 or A549 MAVS$^{-/-}$ cells at 24 hpi. RNA levels were normalized to actin and mock-treated controls of each cell type. Data are presented as mean + s.d. (n = 6 obtained from 3 biological replicates). Statistical significance was assessed by two-way ANOVA with Dunnett's multiple-comparisons test.

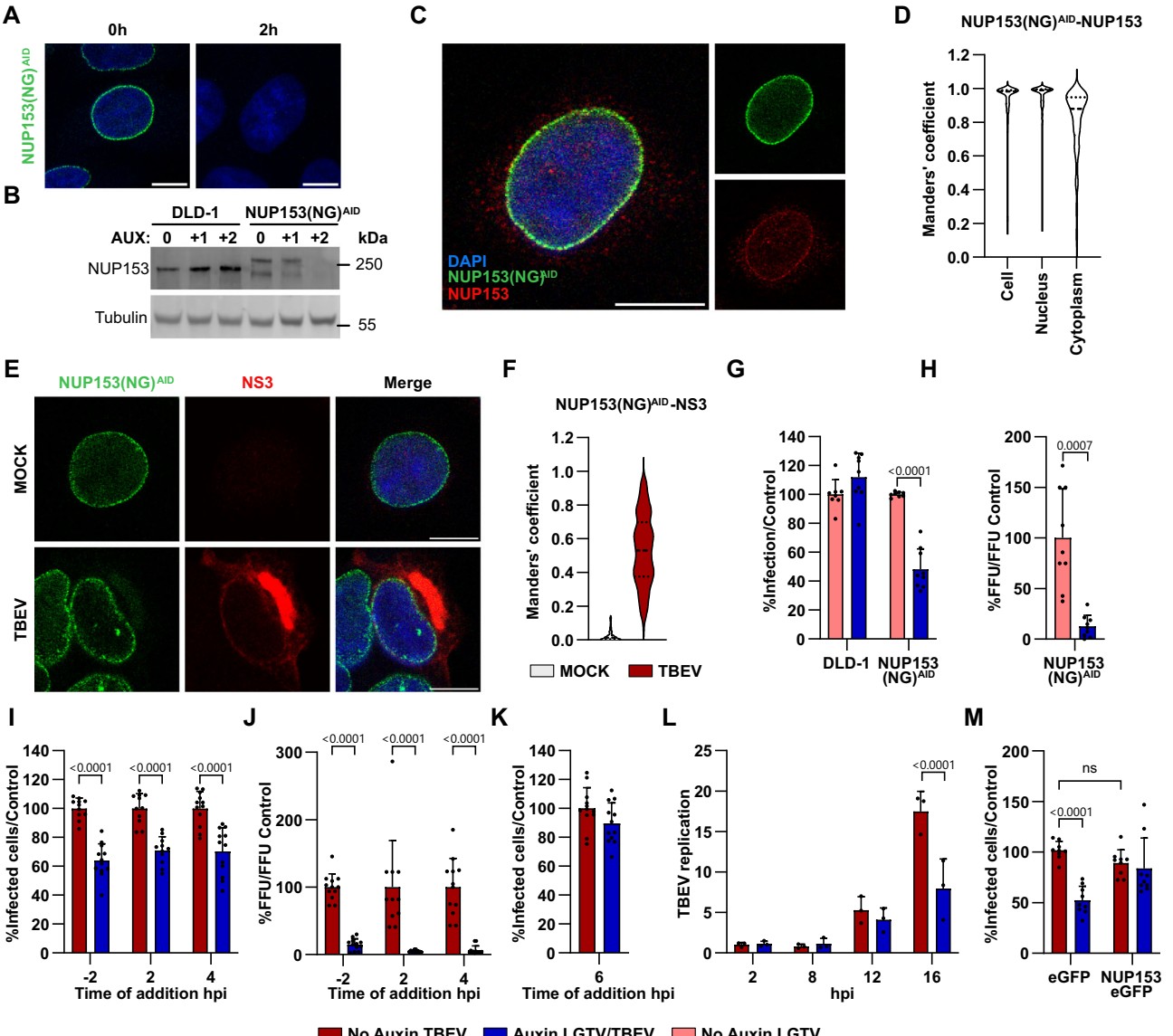

**Fig. 2 | NUP153 acts as a proviral factor during TBEV infection. A**, **B** Auxin-induced degradation of NUP153 in DLD-1 and NUP153(NG)^AID cells treated with 1 mM auxin. **A** Representative immunofluorescence images showing DAPI (blue) and NUP153(NG)^AID (green). Scale bars, 10 µm. **B** Immunoblotting analysis of NUP153 and tubulin. Representative blot from 3 independent experiments.
**C**, **D** Immunofluorescence validation of NUP153(NG)^AID cells stained for DAPI (blue), NUP153(NG)^AID (green) and endogenous NUP153 (red). Scale bar, 10 µm.
**D** Colocalization analysis across cellular compartments using Manders' coefficient (violin plot, n = 111). **E**, **F** NUP153(NG)^AID cells mock-treated or infected with TBEV for 24 hpi. **E** Representative images stained with DAPI (blue), NS3 (red), and NUP153(NG)^AID (green). Scale bars, 10 µm. **F** Quantification of NUP153(NG)^AID–NS3 colocalization in the cytoplasmic segment by using the Manders' coefficient (violin plot; (n = 111 [MOCK] and n = 251 [TBEV]). **G**, **H** Viral infection in DLD-1 and NUP153(NG)^AID cells at 48 hpi with LGTV. **G** Quantification of E-positive cells by plate reader. **H** Released virus measured by focus-forming assay. Signals were

normalized to auxin-uninduced controls (mean + s.d., n = 9; two-tailed Student's t test). **I**, **J** Time-of-addition analysis in NUP153(NG)^AID cells infected with TBEV, with auxin added at indicated times. **I** E-positive cells quantified at 16 hpi. **J** Viral titers in supernatants measured by focus-forming assay. Data normalized to auxin-uninduced controls (mean + s.d., n = 12; two-way ANOVA with Šídák's multiple-comparisons test). **K** Quantification of infected NUP153(NG)^AID cells following auxin treatment at 6 h post infection (mean + s.d., n = 12). **L** TBEV RNA levels measured by RT-qPCR following auxin induction at 1 hpi and collection at the indicated time points. Viral RNA was normalized to actin and auxin-uninduced cells at 2 hpi (mean + s.d., n = 3; two-way ANOVA with Šídák's test). **M** Rescue experiment: E-positive cells quantified at 16 hpi in NUP153(NG)^AID cells transfected with eGFP or eGFP-NUP153 and treated with auxin at −2 hpi. Data were normalized to auxin-uninduced eGFP controls (mean + s.d., n = 9; two-way ANOVA with Šídák's test; ns, not significant). Infection assays (**G**–**M**) represent 3 biological replicates with 1 (**L**), 3 (**G**, **H**, **M**) or 4 (**J**, **K**) technical replicates in each.

replication per se, and that particles released are highly dependent on NUP153 expression.

**TBEV infection affects the NUP153 interactome**
NUP153 thus plays an important role in viral propagation and is recruited to the viral amplification site on the ER during infection. To understand the underlying mechanism, we employed an unbiased proteomic approach to examine the effect of infection on the NUP153 interactome.

Cells transfected with eGFP-NUP153 or eGFP-encoding plasmids were infected with TBEV or mock-treated. The interacting proteins were enriched by protein GFP pulldown and analyzed by mass spectrometry. The analysis revealed subsets of proteins that specifically interacted (directly or indirectly) with NUP153 during either mock treatment or TBEV infection (Fig. 3A), as well as 242 proteins that interacted with NUP153 under both conditions. Pathway analysis (KEGG) of the shared protein set revealed that the NUP153 interactors were involved at

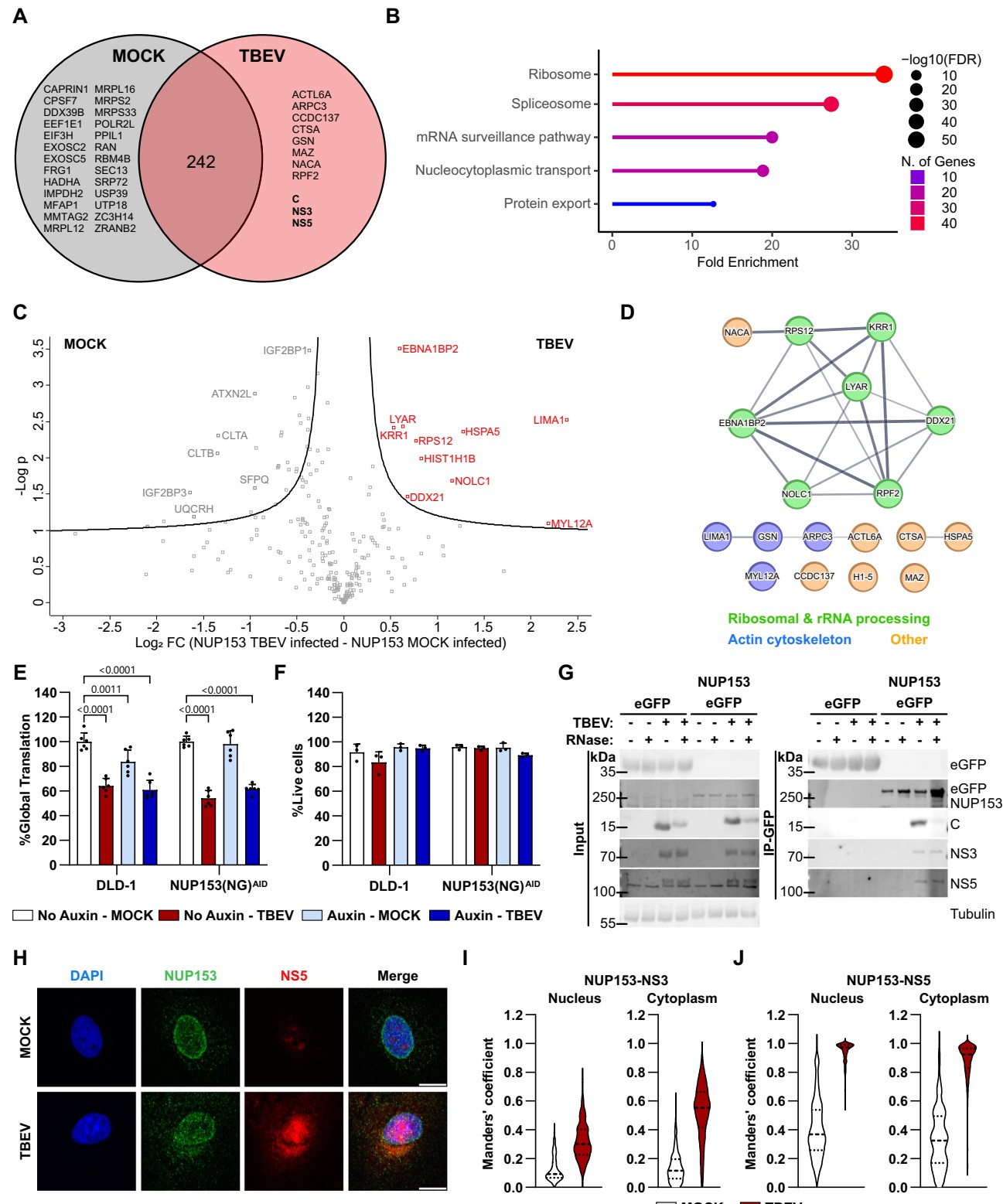

different stages of protein production, from mRNA processing (spliceosome, 37 proteins) to nucleocytoplasmic transport (21 proteins) and translation (ribosome, 47 proteins) (Fig. 3B). Comparison of the data from mock-treated samples with the TBEV infected cells revealed that a set of proteins, involved in gene expression (spliceosome, mRNA surveillance, and ribosome), RNA degradation, and nucleocytoplasmic transport, was either lost or decreased during infection (Supplementary Fig. 3A). Statistical analysis of the 242 proteins found under both

conditions revealed differential enrichment in mock-treated and TBEV-infected samples (Fig. 3C). Notably, during infection, we observed a shift in the cellular interactome of NUP153 towards actin cytoskeleton proteins, as well as a set of ribosomal and rRNA processing proteins (Fig. 3D). Several of these proteins participate in cell cycle regulation (examples EBNA1BP2, LYAR, and RPS12) and are also involved in the translation process (examples DDX21 and NACA), either directly at the ribosome or by regulating ribosomal biogenesis[19–26].

**Fig. 3 | NUP153 associates with ribosomal and rRNA processing proteins, and with TBEV C, NS3, and NS5 proteins during infection. A–D** HEK293T cells expressing eGFP- or eGFP-NUP153 were mock-treated or infected with TBEV. At 24 hpi, lysates were subjected to GFP-trap pulldown followed by mass spectrometry. **A** Venn diagram of NUP153-interacting proteins, identified under mock and TBEV conditions. **B** KEGG pathway enrichment analysis of 242 proteins detected in both conditions (ShinyGO 0.80[77] FDR cutoff 0.05). **C** Volcano plot comparing NUP153 interactomes in mock- and TBEV-infected cells. Statistical significance determined by two-sample Student's $t$ test (FDR 0.05 and $S_0$ 0.1). **D** STRING network of proteins specifically enriched during TBEV infection; edge thickness reflects interaction confidence[78]. **E, F** DLD-1 and NUP153(NG)$^{AID}$ cells were treated ±auxin and mock-treated or infected with TBEV for 48 hpi. **E** Global protein synthesis measured by OPP incorporation ($n = 6$ acquired from 3 biological replicates). **F** Cell viability assessed by trypan blue exclusion ($n = 3$ biological replicates). Data were normalized to auxin-uninduced mock controls and are presented as mean + s.d. Statistical significance was determined by two-way ANOVA with Šídák's multiple-comparisons test. **G** Co-immunoprecipitation from HEK293T cells expressing eGFP or eGFP-NUP153 at 48 hpi with TBEV ($n = 3$ biological replicates). Lysates were treated with or without RNase A prior to GFP-trap pulldown. Inputs and immunoprecipitates were analyzed by immunoblotting for eGFP, C, NS3, NS5 and tubulin. **H–J** Immunofluorescent analysis of A549 cells at 24 hpi with TBEV. **H** Representative image DAPI (blue), NUP153 (green), and NS5 (red). Scale bar 10 μm. Manders' colocalization coefficients were quantified for NUP153–NS3 (**I**) and NUP153–NS5 (**J**) in nuclear and cytoplasmic compartments. Violin plots show individual cells ($n = 560$ [MOCK] and $n = 722$ [TBEV] for NUP153–NS3; $n = 339$ [MOCK] and $n = 360$ [TBEV] for NUP153–NS5).

TBEV and other viruses induce translational shut-off in the host cells[27,28]. We investigated whether NUP153 is involved in this process, as we noted that infection altered its interactions with several proteins that are involved at different stages of gene expression. To quantify the absolute translation levels, we measured the incorporation of O-propargyl puromycin (OPP) into elongating polypeptides in the presence or absence of NUP153 using the AID system. We found that while TBEV infection resulted in a 40% reduction of global translation levels, NUP153 depletion did not affect these levels (Fig. 3E). To verify that the effect we observed was not due to reduced cell viability, we analyzed cell viability using trypan blue following TBEV infection and absence of NUP153 and detected no significant differences (Fig. 3F). These data indicate that infection changes the interactome of NUP153 in the cell and that NUP153 is not involved in the translational shut-off caused by virus infection.

### NUP153 associate with viral protein NS3 and NS5

Approximately 50% of human proteins identified in the NUP153 pull-down experiment were endogenous RNA-binding proteins (Supplementary Table 1). In addition, we identified three RNA-binding viral proteins (C, NS3, and NS5) (Fig. 3A)[29,30]. To determine whether NUP153 requires RNA binding as a bridge to interact with these viral proteins, we performed co-immunoprecipitation assay in the presence or absence of RNase A (Fig. 3G). All three viral proteins co-immunoprecipitated with NUP153 in the presence of RNA, but the interaction between NUP153 and C was reduced after RNase A treatment, indicating that NS3 and NS5 bind to NUP153 independent of RNA, whereas the C interaction is likely RNA dependent. Furthermore, we also find that RNase A treatment resulted in decreased C protein in the input (Fig. 3G), suggesting that RNA could be important for this protein stability. Immunofluorescence analysis revealed that, while the colocalization of NS3 and NUP153 was more pronounced in the cytoplasm than in the nucleus (Figs. 1A and 3I), NUP153 colocalized very strongly with NS5 in the nucleus and cytoplasm during infection (Fig. 3H, J). Next, we transfected cells with constructs encoding FLAG-tagged NS2B3, NS3, or NS5, and used immunofluorescence microscopy to analyze the distribution of these proteins and NUP153 (Supplementary Fig. 3B, C). We observed that the presence of NS3 and NS5 resulted in upregulation of NUP153 levels, while the presence of NS2B3 only resulted in a minimal increase of the NUP153 signal. Accordingly, this could indicate that nuclear localization of the viral proteins might be required for upregulation of expression. Overall, we find that NUP153 interacts with viral proteins NS3 and NS5 in an RNA-independent manner. These interactions could be involved in the upregulation and recruitment of NUP153 to the viral amplification site that we observe during infection.

### The RNA-binding domain of NUP153 is important for NUP153 recruitment to the viral amplification site and binds viral RNA

NUP153 is a large multi-domain protein[31]. The N-terminal domain contains NE-binding site (amino acids 1–144), NPC-targeting domain (amino acids 39–439), and an RNA-binding domain (RBD) (amino acids 250–400). The middle domain contains four zinc fingers, and the C-terminal domain contains FG repeats (Fig. 4A)[31,32]. To investigate which region of NUP153 is important for the recruitment of the protein to the viral amplification site during infection, we transfected HEK293T cells with constructs encoding different eGFP-tagged NUP153 truncation variants, and then infected the cells with LGTV (Fig. 4B). Similar to what was previously reported, in uninfected cells, the variants containing the NE-binding site and NPC-targeting domain (eGFP-NUP153$_{1-650}$, eGFP-NUP153$_{1-250}$ and eGFP-NUP153$_{1-650-RBD}$) correctly localized at the NE, while the variant lacking these domains (eGFP-NUP153$_{880-1475}$ and eGFP-NUP153$_{RBD}$) did not and, instead, were detected diffused in the cytoplasm or in the nucleus respectively (Fig. 4B)[33,34]. Further, the cellular localization of eGFP-NUP153$_{1-650}$, but not eGFP-NUP153$_{1-250}$, eGFP-NUP153$_{1-650-RBD}$ and eGFP-NUP153$_{RBD}$, changed during infection (Fig. 4B), indicating that the RNA-binding domain is required but not sufficient for the recruitment of NUP153 to the viral amplification site.

Next, we performed a crosslinking immunoprecipitation (CLIP) experiment to investigate whether NUP153 can interact with viral RNA (Fig. 4C). First, protein expression of the NUP153-GFP constructs were verified by WB (Supplementary Fig. 4A). As a control we verified that NUP153 yet not eGFP was able to bind cellular RNA encoding for actin (Supplementary Fig. 4B). The analysis of the interaction with viral RNA revealed that full-length NUP153, as well as the N-terminal (eGFP-NUP153$_{1-650}$), the construct only expressing the RBD (eGPF-NUP153$_{RBD}$) and C-terminal (eGFP-NUP153$_{880-1475}$) variants can bind viral RNA (Fig. 4C). As the FG repeat-containing C-terminal domain of NUP153 binds multiple proteins, including viral proteins (HIV), the interaction with viral RNA may have been mediated by indirect interaction involving additional binding partners[35]. The eGFP-NUP153$_{1-250}$ and the eGFP-NUP153$_{1-650-RBD}$ variants, which both lack the RNA-binding domain, did not bind viral RNA, suggesting that the RNA-binding region[36] is required for viral RNA binding (Fig. 4C). We also investigated whether the interaction between NUP153 and viral RNA is RNA polarity-specific. We observed that NUP153 bound both positive- and negative-strand viral RNA, although the interaction with the positive-strand was more pronounced than that with the negative-strand (Fig. 4D).

Furthermore, we performed RNA electrophoretic mobility shift assay (RNA EMSA) using purified RBD domain of NUP153 (His-eGFP-NUP153$_{RBD}$) and in vitro transcribed viral TBEV genome to confirm that the RBD domain alone could interact with viral RNA without other cellular components. The interaction with TBEV viral RNA was confirmed, finding a shift at -14 μM (Fig. 4E).

The RBD is thus responsible for the viral RNA interaction, and the N-terminal domain 1–650 is important for the redistribution of NUP153 to the ER during infection. However, it is unclear which domains are important for viral infection. Therefore, NUP153(NG)$^{AID}$ cells were transfected with the different NUP153 mutants in the presence or absence of auxin and infected with TBEV. We found that only the full

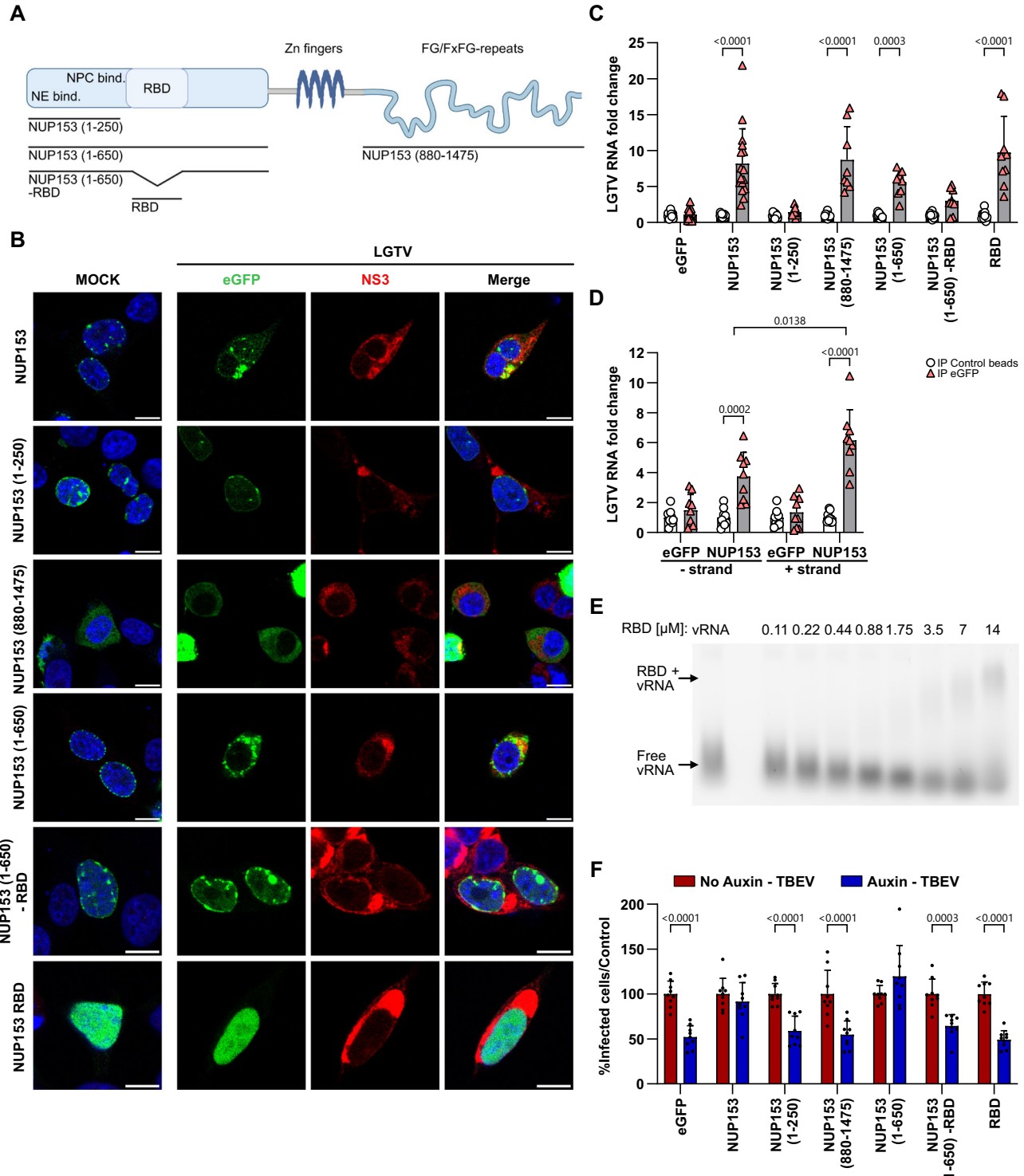

**Fig. 4 | The RNA-binding domain of NUP153 is required for infection, NUP153 recruitment to the viral amplification site and binding of viral RNA.**
**A** Schematic representation of full-length NUP153 and truncation variants: NUP153$_{1-250}$, NUP153$_{1-650}$, NUP153$_{880-1475}$, NUP153$_{1-650-RBD}$ and RBD alone. NE nuclear envelope, NPC nuclear pore complex, RBD RNA-binding domain.
**B** Immunofluorescence analysis of HEK293T cells expressing eGFP-NUP153 or truncation variants and infected with LGTV for 48 h. Representative confocal images show DAPI (blue) and NS3 (red), and eGFP (green). Scale bars, 10 μm.
**C, D** Crosslinking immunoprecipitation (CLIP) in HEK293T cells expressing eGFP-NUP153 or truncation variants at 48 hpi with LGTV. cDNA was generated using random primers (**C**) or strand-specific viral primers (**D**). Immunoprecipitated viral RNA was quantified by RT-qPCR and normalized to input positive or negative-

strand RNA and to control bead samples. Data are shown as mean + s.d. ($n = 9$ acquired from 3 biological replicates). Statistical significance was assessed by two-way ANOVA with Šídák's multiple-comparisons test. **E** RNA electrophoretic mobility shift assay (RNA EMSA) using Cy3-labeled full-length TBEV RNA incubated with increasing concentration of purified His-eGFP-NUP153$_{RBD}$. **F** Functional complementation assay. NUP153(NG)$^{AID}$ cells were transfected with eGFP, eGFP-NUP153 or truncation variants, treated with auxin at −2 hpi and infected with TBEV. E-positive cells were quantified at 16 hpi by plate reader. Data were normalized to auxin-uninduced controls and are presented as mean + s.d. ($n = 9$ acquired from 3 biological replicates). Statistical significance was determined by two-way ANOVA with Šídák's multiple-comparisons test.

length and the NUP153$_{1-650}$ could rescue the infection (Fig. 4F). Because the RBD domain alone was not able to rescue infection, we hypothesize that the remaining N-terminal domain is required for recruitment to the replication site and interaction with the viral proteins. To investigate this, we performed a co-immunoprecipitation assay with the N-terminal (eGFP-NUP153$_{1-650}$) and C-terminal (eGFP-NUP153$_{880-1475}$) domains of NUP153 and found that the N-terminal domain is required for the interaction with viral protein NS3 (Supplementary Fig. 4C).

Together, these observations indicated that the RBD of NUP153 is required for binding to viral RNA. However, the entirety of the N-terminal domain of NUP153 (1–650aa) is required for interaction with viral protein NS3, recruitment to the viral amplification site and the proviral function.

## NUP153 binds two conserved regions in the viral genome

To determine if there was a specific site on the viral RNA that interacted with NUP153, we analyzed cells transfected with eGFP-NUP153$_{1-650}$ encoding construct and infected with TBEV, using CLIP-sequencing (CLIP-seq) (Fig. 5A). We opted to use the eGFP-NUP153$_{1-650}$ instead of the full-length construct to minimize indirect RNA binding arising from proteins bound to the FG-domain of NUP153 (Supplementary Fig. 4D). NUP153 primarily bound host RNA, however, approximately 1.4% of reads mapped to the TBEV genome (Fig. 5B), particularly the positive strand (Fig. 5C). Statistical analysis of the positive-strand reads revealed association with three viral RNA regions, depicted as peaks in Fig. 5C. The first peak was located at the end of the *E* protein coding region (peak 1, 2118–2264 nt), a region that is highly conserved among orthoflaviviruses, while the two other peaks (peak 2, 5760–5948 nt; and peak 3, 6950–6970 nt) were located in the *NS3* and *NS4B* coding sequences, respectively (Fig. 5C and Supplementary Fig. 4E–G).

The RNA-binding domain of NUP153 has previously been shown to associate with G- and GC-rich regions of RNA molecule[11,36]. G-rich sequences can fold into G-quadruplex (G4) structures, a type of four-stranded nucleic acids secondary structure that can fold in certain G-rich sequences[37]. Peak 1 contains two predicted G4 structures (Fig. 5D boxes and Supplementary Fig. 6A). The second of which is highly conserved among orthoflaviviruses sequences (Supplementary Fig. 4E)[38,39]. We used in vitro transcription to generate RNA fragments containing the wildtype (WT) peak 1, a mutated construct (11G1C) lacking the second highly conserved G4 motif and a construct lacking all orthoflavivirus conserved Gs (28G) (Fig. 5D). We also in vitro transcribed three versions of peak 2, the WT peak, a mutated construct lacking the GA pattern that we identified to be conserved among orthoflaviviruses (8G8A) and a construct lacking all conserved Gs (19G) (Supplementary Fig. 5A). RNA-EMSA was used to determine the interaction of the NUP153 RBD with the different RNA fragments corresponding to peak 1 and 2 and their mutants. We found that WT fragments of peaks 1 and 2 bound better to the NUP153 RBD compared to the 28G and 19G fragments, respectively, that lacked all conserved Gs (Fig. 5E–H and Supplementary Fig. 5B, C), indicating that Gs in these regions are important for the interaction with NUP153. Additionally, we found that removing the conserved GA pattern (8G8A) in peak 2 does not affect binding to the RBD, suggesting that it is not important for the interaction. These results showed that NUP153 bind specific regions in the TBEV genome, which contains conserved Gs among orthoflaviviruses, and these are important for effective binding.

## NUP153 interacts with a conserved G-rich region on viral RNA if the G-quadruplex (G4) structure is destabilized

In peak 1, we predicted one highly conserved G4 motif among all orthoflaviviruses, however we do not know whether this region can indeed fold into G4s and if this motif is required for interaction with NUP153. To test if peak 1 can fold into G4 we generate longer in vitro

transcribed fragments of the wildtype (WT) peak and a mutated construct (11G1C) lacking the second, highly conserved G4 (Supplementary Fig. 6B) and transfected these into cells and measured their ability to form G4s, by quantifying the mean intensity signal after detection with anti-G4 antibody (1H6). We detected G4 signal from both transfections, however, the signal intensity of WT RNA was significantly higher compared to 11G1C RNA, suggesting that the WT sequence has higher propensity to fold into stable G4 structures (Fig. 6A, B). To verify the existence of the two different G4s, we then in vitro transcribed the WT and mutant versions of peak 1 and analyzed the properties using circular dichroism (CD) spectroscopy (Fig. 6C, D). G4s formed in RNA oligonucleotides are reported to predominantly form parallel G4s, with positive maximum around 264 nm, and their stabilization is dependent on potassium ions[40]. The CD analyses showed a positive maximum at 264 nm for the WT and the 11G1C mutant peak 1 in the presence of KCl, while in its absence, the maximum shifted to 268 nm for the WT and 272 nm for the 11G1C (Fig. 6C). These data suggested that both the WT sequence and the mutant 11G1C may form parallel G4 structures in the presence of KCl. The small shift in the positive maximum of the WT fragment in the absence of KCl could be explained by the formation of complex secondary structures along the RNA molecule, including RNA stem-loop structures, which produces overlapping CD spectra with RNA G4s[41,42]. In contrast, the 28G derivative, which lacked all (28) conserved guanines (including the two G4s) but contained 17 none-conserved guanines showed a maximum at 272 nm in all conditions (Fig. 6C) indicating that 28G mutant cannot form RNA G4 secondary structures. To further confirm the effect of KCl in G4 formation and stabilization in these sequences, we conducted thermal stability analysis using temperature-induced CD melting experiments. These experiments showed that the melting temperatures ($T_m$) of both WT and 11G1C sequences were significantly increased upon KCl addition by around 3 and 6 °C, respectively, while no change in $T_m$ was observed for 28G mutant in presence of KCl (Fig. 6D).

Even though the analysis indicated that peak 1 can form a G4, it was not clear whether the G4 was bound by NUP153. In vitro, several sequences are able to form G4s, yet most of these are found unfolded in vivo, potentially through a complex mechanism involving proteins that bind G-rich RNA regions[40]. We therefore quantified eGFP-NUP153$_{1-650}$ binding to the different RNA fragments in vitro in the presence or absence of KCl. We found that eGFP-NUP153$_{1-650}$ bound significantly more WT RNA in the absence of KCl than in the presence of KCl (Fig. 6E). Similarly, the absence of KCl improved the eGFP-NUP153$_{1-650}$ binding to the 11G1C derivative, which lacked the second G4 motif. In contrast, only low levels of binding were observed with the 28G derivative. This interaction was not significantly affected by KCl (Fig. 6E). This indicated that peak 1 form G4 motifs in the presence of KCl, but NUP153 interacted with peak 1 in the absence of G4 secondary structure.

Similar to peak 1, peak 2 (in the *NS3* coding sequence) is conserved among orthoflaviviruses and contains a predicted G4 structure (Supplementary Fig. 6A)[37]; however, the G4 is not conserved (Supplementary Fig. 4F). The total percentage of guanines in peak 2 is similar to that in peak 1 (31% vs. 32%) (Supplementary Fig. 6C) but, while 56% of these residues are conserved in peak 1, only 31% are conserved in peak 2. No conserved guanines were located in peak 3 (Supplementary Fig. 4G). To investigate whether the conserved guanines or GAs identified in peak 2 are important for the binding to NUP153, we performed an in vitro binding assay with WT and mutated RNA sequences. We found that in the presence of KCl, peak 2 WT, 8G8A, and 19G RNA derivatives (Supplementary Fig. 5A) bound eGFP-NUP153$_{1-650}$ to a similar degree as peak 1 WT in the absence of KCl (Fig. 6E, F). This indicated that the binding of NUP153 to peak 2 was not mediated by the conserved guanines or GAs, which is in line with the RNA EMSA results in Fig. 5G.

Taken together, the above findings indicated that NUP153 binds positive-stranded RNA, and specifically, conserved guanines in peak 1, in the absence of a G4.

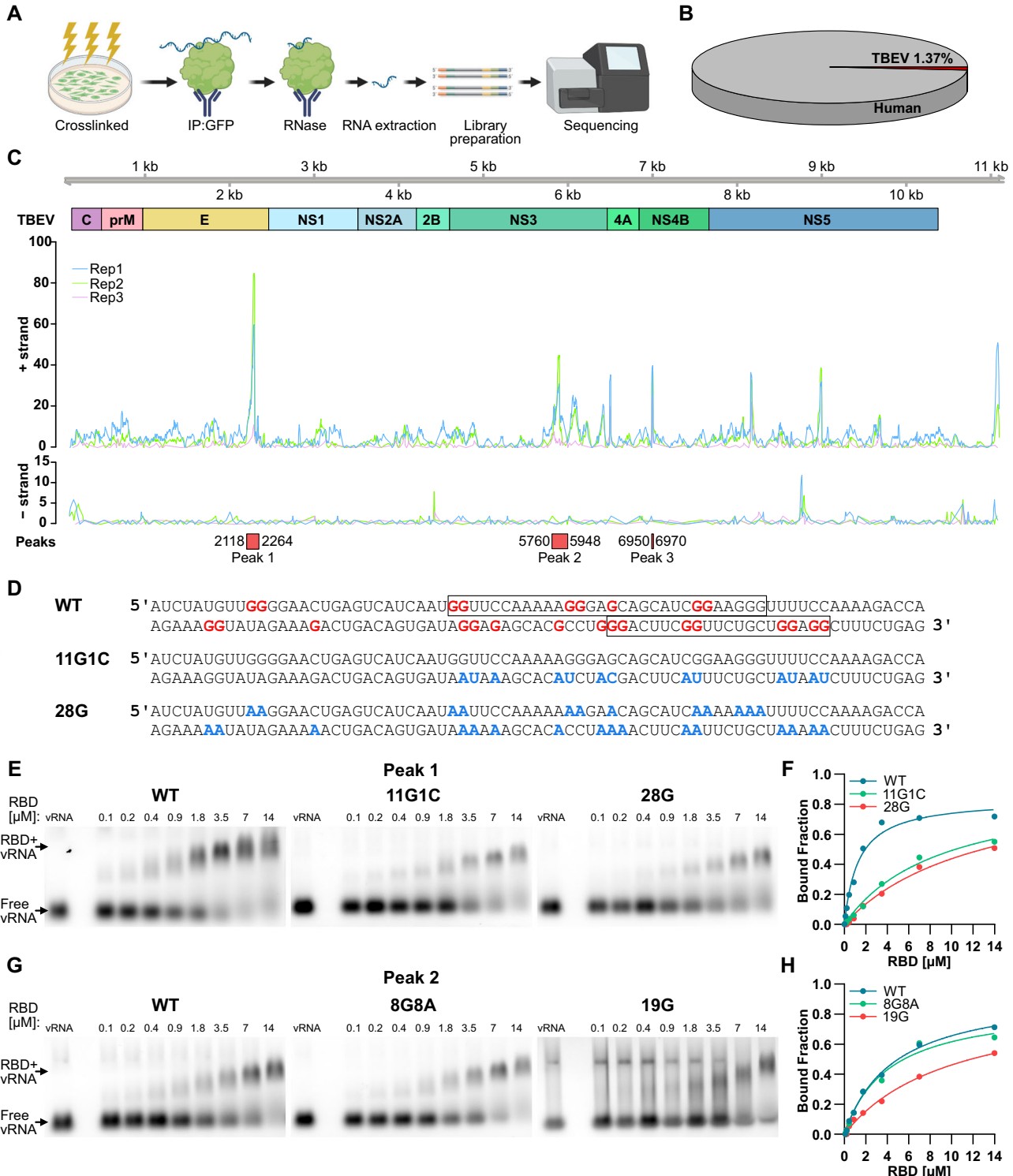

**Fig. 5 | NUP153 interacts with viral RNA at a conserved G site at the end of *E* coding sequence. A** Schematic of the CLIP-seq workflow. HEK293T cells expressing eGFP-NUP153$_{1-650}$, were infected with TBEV, and subjected to CLIP-seq at 48 hpi. **B** Quantification of CLIP-seq reads mapping to TBEV RNA relative to human transcripts. **C** Genome-wide distribution of positive- and negative-strand CLIP-seq reads mapped to the TBEV genome (three biological replicates). Enriched positive-strand binding peaks are highlighted in red boxes. **D** Sequences corresponding to peak 1 (WT, wildtype), and mutant derivatives (11G1C and 28G). Orthoflavivirus-conserved guanines are indicated in red and mutations in blue. Predicted G4s motifs are boxed[37]. **E**–**H** RNA EMSA using Cy3 labeled RNA fragments corresponding to peak 1 [2118−2264] (**E**) and peak 2 [5760−5948] (**G**), incubated with increasing concentrations of purified His-eGFP-NUP153$_{RBD}$ protein. Binding curves derived from RNA EMSA are shown in (**F**, **H**).

## NUP153 is a proviral factor during orthoflavivirus infection

Since the guanines needed for NUP153 binding in peak 1 are conserved among all orthoflaviviruses, we next sought to investigate the importance of NUP153 during infection with other prominent orthoflaviviruses. We first evaluated the intracellular localization of NUP153 during infection with ZIKV, JEV, WNV, YFV, and DENV2. We found that, similar to TBEV and LGTV infection, infection with other orthoflaviviruses induced the expression of NUP153 and recruited

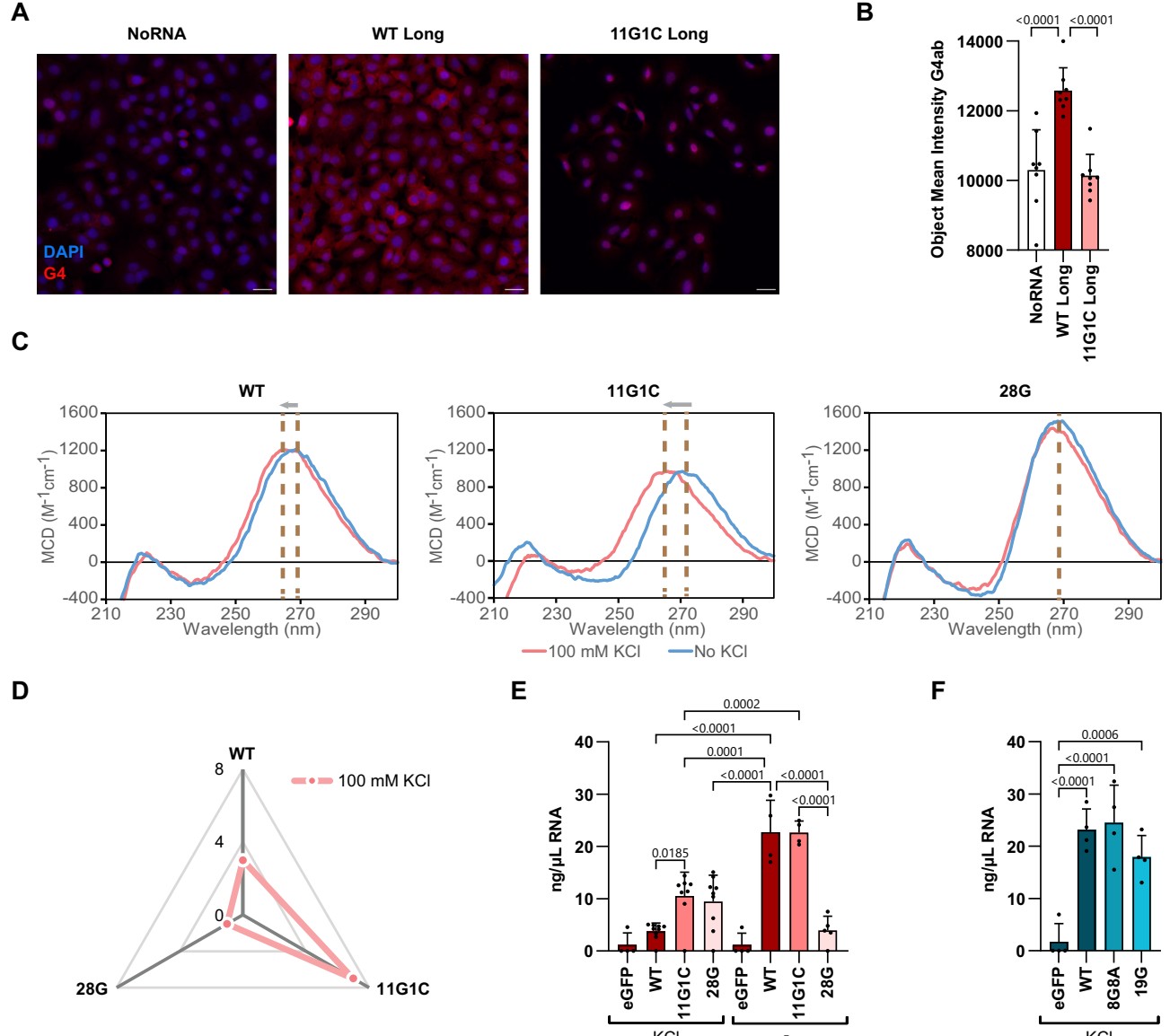

**Fig. 6 | NUP153 interacts with viral RNA in the absence of G4 motif.**
**A**, **B** Immunofluorescence analysis of A549 cells transfected with viral RNA fragments (WT long-peak 1 and 11G1C long-peak 1) and stained for DAPI (blue) and G4 structures (red). Images were acquired using a Cytation5 system. Scale bars, 30 μm. **B** Quantification of anti-G4 mean fluorescence intensity per cell. Data are presented as mean + s.d. ($n = 8$ acquired images from 4 biological replicates). Statistical significance was determined by one-way ANOVA with Fisher's LSD test. **C** Circular dichroism (CD) spectral of WT, 11G1C, and 28G fragments in the presence of KCl at 25 °C. Arrows indicate peak shift in the presence of KCl. **D** Spider plot

showing changes in melting temperature ($\Delta T_m$) for WT, 11G1C, and 28G fragments in the presence versus absence of KCl, determined by CD melting analysis (25–95 °C temperature range, 250–285 nm wavelength range). In vitro RNA-binding assays using purified eGFP-NUP153$_{1-650}$ incubated with mutant RNA fragments derived from peak 1 (**E**) or peak 2 (**F**), in the presence of KCl. Bound RNA was quantified by Qubit HS RNA assay. eGFP incubated with WT RNA served as a negative control. Data are shown as mean + s.d. ($n = 4$ biological replicates). Statistical significance was assessed by one-way ANOVA with Šídák's multiple-comparisons test.

NUP153 to the cytosolic fraction (Fig. 7A, B). Furthermore, depletion of NUP153 using the AID-system resulted in a reduced number of infected cells in all the tested orthoflaviviruses and a strong reduction in progeny particles released (Fig. 7C–E), indicating that NUP153 is an important conserved host protein in the orthoflavivirus life cycle.

**NUP153 plays an important role in flavivirus infection by impacting the ratio of structural to nonstructural viral proteins**
Orthoflaviviruses have a single-stranded RNA genome that encodes a polyprotein that is subsequently processed into 10 viral proteins. Many viruses employ mechanisms to regulate viral protein translation, increasing the translation of specific viral proteins while decreasing the translation of other proteins to ensure optimal replication. However,

how orthoflavivirus protein is regulated; during translation or via post translational modification and whether all proteins are expressed equally during different stages of the viral amplification cycle is unknown[43]. Since we found that NUP153 interacts with numerous ribosomal proteins (Supplementary Table 1), is required early on during infection (Fig. 2I, J), and interacts with the viral mRNA at the end of the coding region of the structural proteins (Fig. 5C, E), we hypothesized that it could play a major role in regulating viral RNA translation to increase the ratio of viral structural proteins compared to nonstructural proteins, leading to enhanced viral particle production during the early stages of infection. To test this hypothesis, we performed an in vitro translation assay of the TBEV genome in the presence or absence of purified NUP153 RBD. We used a mRNA construct

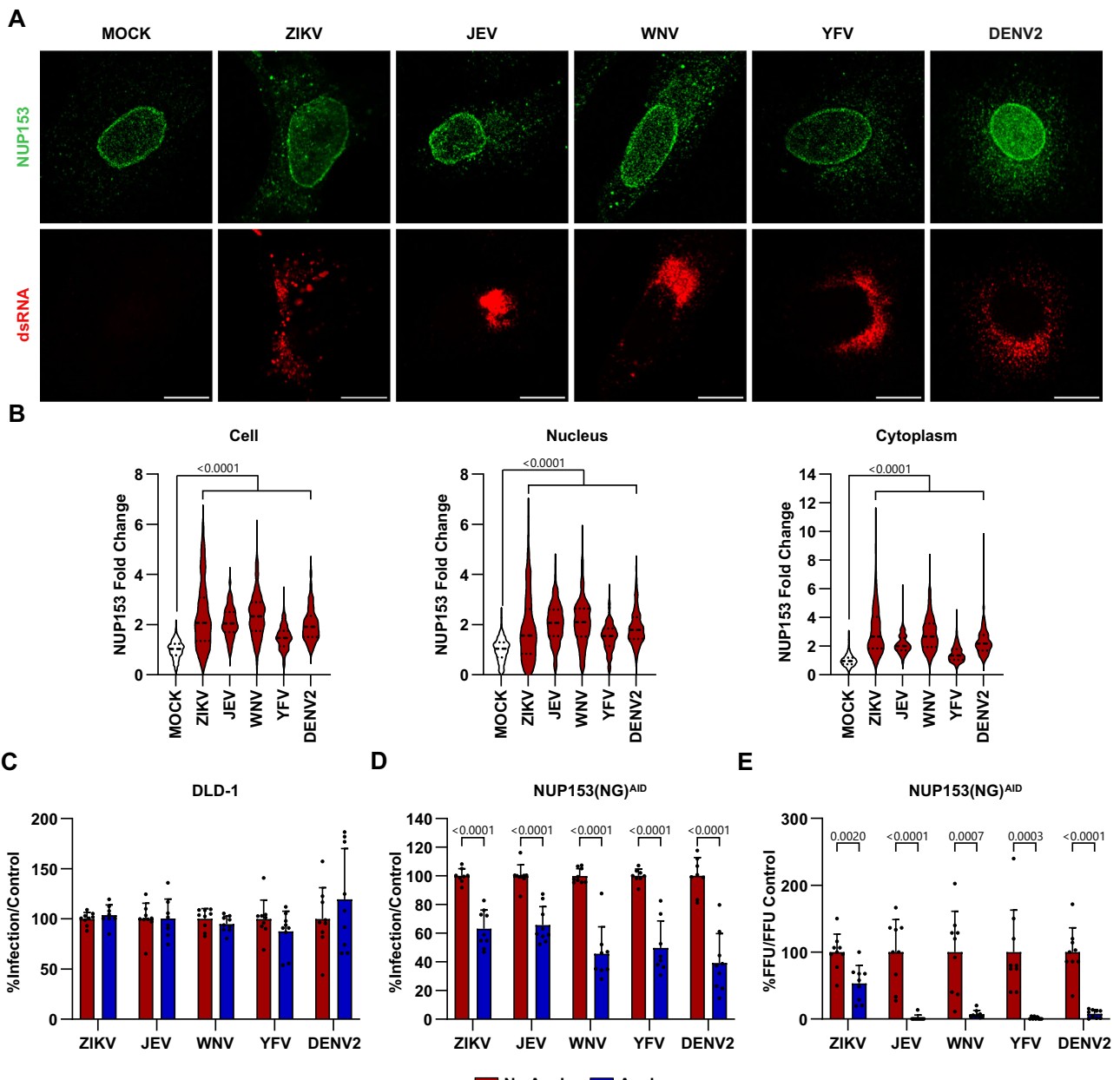

**Fig. 7 | NUP153 is recruited to the replication site and important for infection during orthoflavivirus infection. A**, **B** Immunofluorescence analysis of A549 cells infected with ZIKV, JEV, WNV, or YFV for 24 h, or DENV2 for 48 h. **A** Representative confocal images of cells stained for dsRNA (red) and NUP153 (green). Scale bars, 10 μm. **B** NUP153 integrated intensity in whole cells, nuclear and cytoplasmic compartments, expressed as fold change relative to mock. Violin plots show individual cells ($n = 574$ [MOCK], $n = 236$ [ZIKV], $n = 239$ [JEV], $n = 216$ [WNV], $n = 193$ [YFV], $n = 209$ [DENV2]). Statistical significance was determined by one-way ANOVA with the Kruskal–Wallis test. **C**–**E** DLD-1 or NUP153(NG)^AID cells were treated with auxin or left untreated 2 h prior to infection with ZIKV, JEV, WNV, YFV, or DENV2 and analyzed at 48 hpi. **C**, **D** Quantification of E-positive cells. **E** Viral titers from NUP153(NG)^AID cells measured by focus-forming assay. Data were normalized to auxin-uninduced controls and are presented as mean + s.d. ($n = 9$ acquired from 3 biological replicates). Statistical significance was assessed by two-tailed Student's $t$ test.

that encodes for the 5′ untranslated region, the viral proteins C, prM, and E, together with luciferase (Fig. 8A and Supplementary Fig. 7A). As a control we also used an mRNA in which the binding peak in E was mutated, however as the terminal part of E containing the binding peak can be found duplicated after the luciferase (Mut-28G), we expect this mutated construct to still bind the RBD of NUP153 (Fig. 8A and Supplementary Fig. 7A). The binding of RBD to both these constructs was first confirmed by RNA EMSA, with the WT construct having a higher binding affinity compared to the Mut-28G mutant (Fig. 8B, C, and Supplementary Fig. 7B, C). These constructs were then used for in vitro translation in the presence of either purified His-eGFP-NUP153$_{RBD}$ or

His-eGFP protein. We found that the addition of the NUP153 RBD resulted in a 50% reduction in the luciferase signal compared to addition of eGFP for the WT mRNA construct, while resulting in an increase in luciferase signal for the Mut-28G construct (Fig. 8D). To further investigate the translation products produced during these reactions we performed Western blot using anti-TBEV E antibody to detect in vitro translated proteins (Supplementary Fig. 7D). We confirm that the addition of NUP153 RBD results in a reduction of the WT full length luciferase construct, however we still observe translation of a lower molecular weight construct that could correspond to a translation product encoding for C-prM-E lacking the luciferase.

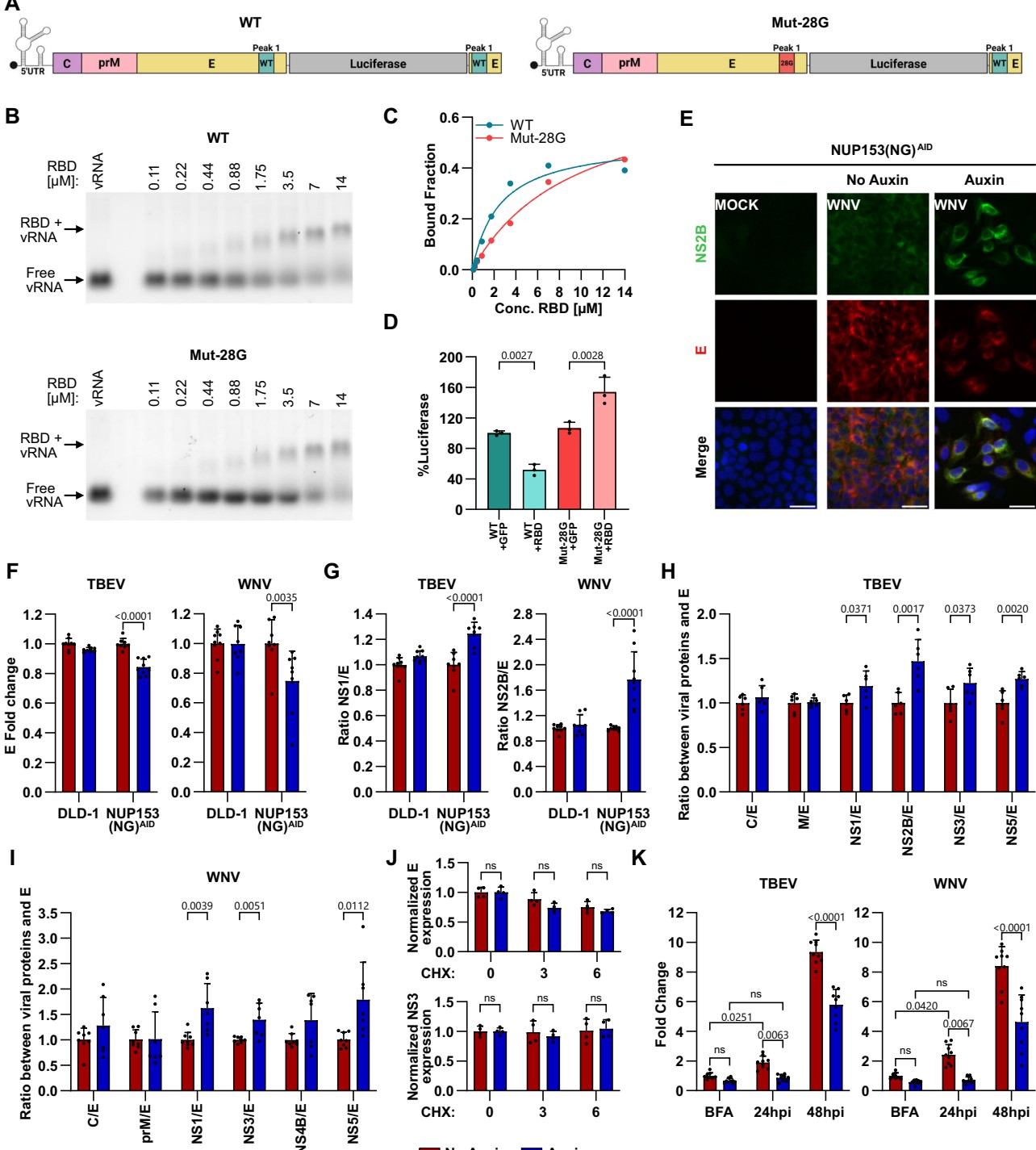

**Fig. 8 | NUP153 regulates the viral structural to nonstructural protein ratio during orthoflavivirus infection. A** Schematic of mRNA constructs used for in vitro translation. **B**, **C** RNA EMSA of Cy3-labeled wild-type or mutated RNA fragments incubated with increasing concentrations of purified His-eGFP-NUP153$_{RBD}$ protein. **C** Binding curves derived from RNA EMSA experiments. **D** In vitro translation assays using constructs in (**A**) in the presence of His-eGFP-NUP153RBD or His-eGFP (4 μM). Luciferase activity was normalized to WT + His-eGFP control (mean + s.d., $n$ = 3 biological replicates; one-way ANOVA with Tukey's test). **E–G** Viral protein expression in DLD-1 and NUP153(NG)$^{AID}$ cells treated ±auxin 2 h before infection with TBEV (16 hpi) or WNV (48 hpi). **E** Representative images of WNV-infected NUP153(NG)$^{AID}$ cells stained for DAPI (blue), NS2B (green), and E (red). Scale bars, 30 μm. **F** Quantification of E intensity normalized to uninduced controls. **G** Ratios of NS1 (TBEV) or NS2B (WNV) to E signal in E-positive cells,

normalized to uninduced samples (mean + s.d., $n$ = 9 acquired from 3 biological replicates; two-way ANOVA with Šídák's test). Viral protein ratios in NUP153(NG)$^{AID}$ cells ±auxin following TBEV (24 hpi) and WNV (48 hpi) infection, quantified using Western blot (**H**) or mass spectrometry (**I**), respectively. Protein ratios were normalized to uninduced controls (mean + s.d.; TBEV $n$ = 6, WNV $n$ = 8; two-tailed Student's $t$ test. **J** Stability of E and NS3 in TBEV-infected NUP153(NG)$^{AID}$ cells ±auxin treated with cycloheximide for 0–6 h at 24 hpi. Protein levels were normalized to tubulin and to 0 h uniduced controls (mean + s.d., $n$ = 4 biological replicates; two-way ANOVA with Šídák's test; ns, not significant). **K** Viral spread in NUP153(NG)$^{AID}$ ±auxin prior to TBEV or WNV infection. Cells were treated with brefeldin A at 16 hpi. E-positive cells were quantified and normalized to BFA-treated uninduced controls (mean + s.d., $n$ = 9 acquired from 3 biological replicates; two-way ANOVA with Šídák's test; ns).

Interestingly, for the Mut-28G construct, we also observed enhanced production of the lower molecular weight product (C-prM-E) in the absence of NUP153 RBD, suggesting that this sequence might also be important for efficient polyprotein translation. However, addition of the NUP153 RBD to the Mut-28G mRNA stabilized translation of the full-length construct in line what we also observed with measurement of the luciferase (Supplementary Fig. 7D and Fig. 8D). Taken together our results indicate that the NUP153 RBD can regulate translation in vitro through binding of a specific site at the end of the E coding region of the viral mRNA.

To determine whether this translation regulation could also be observed in infected cells, we measured the levels of E and NS1 in TBEV-infected cells, and E and NS2B in WNV-infected cells using the AID-system and immunostaining. We observed that NUP153 depletion resulted in an approximately 20% reduction in E levels in E positive cells for both, TBEV and WNV (Fig. 8E, F). However, the expression of TBEV NS1 and WNV NS2B increased in the absence of NUP153 (Supplementary Fig. 8A, B), leading to higher NS1/E and NS2B/E ratios (Fig. 8G), whereas no difference in protein expression was detected in control DLD-1 cells (Fig. 8G). To further investigate the differences in the expression of the different viral proteins in the presence or absence of NUP153 we decided to measure the ratio of E to other TBEV viral proteins using Western blots (Supplementary Fig. 8C). We found that depletion of NUP153 resulted in 40–60% reduction of the different viral proteins (Supplementary Fig. 8D). Additionally, when looking at the ratios between E and the other viral proteins, we observed that similar to our immunofluorescence assay (Fig. 8G), depletion of NUP153 resulted in an increased ratio of nonstructural proteins (NS1, NS2B, NS3, and NS5) to E, while no difference was observed for the ratios of the other structural protein (C and M) (Fig. 8H). To validate our findings using a different orthoflavivirus, we decided to also measure the ratios and viral protein levels in WNV infected cells, however, as we lack antibodies for this virus, we instead used mass spectrometry to detect these. Using mass spectrometry we were able to detect seven of the viral proteins (C, prM, E, NS1, NS3, NS4B, and NS5), with the levels of these significantly being reduced when NUP153 was depleted (Supplementary Fig. 8E). Next, we determined the protein ratio between structural and nonstructural proteins in the same samples. We found no difference in the protein ratio between the structural proteins (C/E and prM/E) and E in the presence or absence of NUP153 (Fig. 8I). However, similar to what we observed for TBEV, the protein ratio of NS1/E, NS3/E and NS5/E showed an increase after auxin treatment and depletion of NUP153 (Fig. 8I) supporting the immunofluorescence quantification of WNV NS2B/E ratio (Fig. 8G), indicating that this is a general trend of the E versus nonstructural proteins. Furthermore, to exclude that the effects we observed was a result of decreased protein stability following NUP153 depletion, we performed a cycloheximide (CHX) assay on TBEV-infected cells and found no significant differences in the stability of the viral proteins E, NS2B and NS3 between auxin-treated and untreated samples over time (Fig. 8J and Supplementary Fig. 8F, G). However, we did observe a small change in NS1 expression 6 h post CHX treatment (Supplementary Fig. 8F, G). Taken together, this indicated that NUP153 interacts with the RNA corresponding to the E terminal region, and thereby modulates the expression of downstream nonstructural proteins to favor the expression of the structural proteins.

Efficient production of progeny virions early in infection is crucial for viral spread and for preventing the cellular antiviral type 1 interferon signaling leading to enhanced infection[44–46]. Accordingly, we measured the efficiency with which the TBEV and WNV viruses spread to new cells in the presence or absence of NUP153. Brefeldin-A (BFA), which disassembles the Golgi complex and prevents virus release[47], was used as a reference of initial infection, and was added at 16 hpi. In the presence of NUP153, both viruses spread to new cells between 16 hpi (BFA was added) and 24 hpi. However, in the absence of NUP153,

no additional cells became infected in the first 24 hpi for both TBEV and WNV, compared to the BFA-treated cells (Fig. 8K and Supplementary Fig. 8H, I). At 48 hpi, many new cells become infected even in the absence of NUP153, indicating that NUP153 is more important early in infection.

Taken together, our results suggest that NUP153 binds to orthoflaviviral RNA at the end of the E coding region, thereby balancing the protein ratio in favor of the structural proteins, leading to increased virion production and release. In the absence of NUP153, the viral protein ratios changes, decreasing the structural protein levels in the cells and thereby delaying the virion release (Fig. 9).

## Discussion

Our results provide a detailed insight into the molecular mechanism of orthoflaviviral co-opting of NUP153 to modulate the ratio of structural to nonstructural protein to promote particle release. We show that NUP153 is recruited to the virus amplification site, and provide data supporting a model in which NUP153 directly interacts with the viral proteins NS3 and NS5, and with a G-rich, highly conserved region of viral RNA to shift the ratio in favor of viral structural proteins production and effective spread of the virus. Importantly, the proposed mechanism is conserved among mosquito and tick-borne orthoflaviviruses.

Most nucleoporins are long-lived and essential for the viability of the cell, making them hard to target with conventional RNAi or CRISPR methods. Therefore, identification and characterization of their importance in viral infection is rare. The AID system allows rapid degradation of a protein of choice in a control system induced conditionally by the addition of auxin[18], enabling the study of NUP153 in virus infection. Our data indicates that NUP153 interacts with orthoflavivirus proteins NS3 and NS5 in the ER at the viral amplification site during infection (Fig. 3G–J). In a previous study, ZIKV and DENV NS2B3 have been reported to degrade FG nucleoporins, including NUP62, NUP98, and NUP153[48]. Contrary to this, the data presented in the current study suggest that TBEV NS3 and NS5 might be involved in upregulation of NUP153 expression (Supplementary Fig. 3B, C) and that NS5 is involved in the recruitment of NUP153 to the viral amplification site during infection. Similar observations have been made for HCV, where NUP153, together with other NUPs, are upregulated upon infection and recruited to the membranous web sites were replication and assembly steps take place[13]. Additionally, knock-down of NUP98, NUP153 and NUP155 results in a reduced HCV infection, similar to what we observed for orthoflavivirus infection, indicating potential similarities in the proviral mechanism of NUP153 for members of the Flaviviridae family[13].

Neufeldt et al.[13] showed that the NUP153, as well as other NUPs, are important for the generation of the membranous web induced by HCV. They suggested that NUPs are important for the sorting of viral proteins to ensure that only NLS-containing proteins would have access to the replication site and exclude pattern recognition receptors from mounting an interferon response[13,49]. Orthoflaviviruses, on the other hand, replicate within small vesicular-like structures[50,51] and, although we cannot exclude the possibility that additional NUPs are also recruited to the ER during orthoflavivirus infection[52], we did detect NUP153 and NS3 colocalization, with only a small fraction of NUP153 colocalizing with dsRNA at the replication vesicle (Fig. 1D–G). Additionally, we primarily detected positive-stranded RNA–NUP153 interaction (Fig. 5C), indicating collectively that NUP153 is not directly involved in virus replication within the replication vesicles.

We showed that NUP153 interacts with a conserved G-rich region on viral RNA and is important for viral infection. While this region can potentially form G4, this structure is not required for NUP153 binding and the binding is reduced in vitro in the presence of potassium, which is needed for G4 formation and stabilization. Mammalian cells contain some RNA sequences that can form G4s in vitro but remain unfolded in vivo, through a complex mechanism that has not been fully

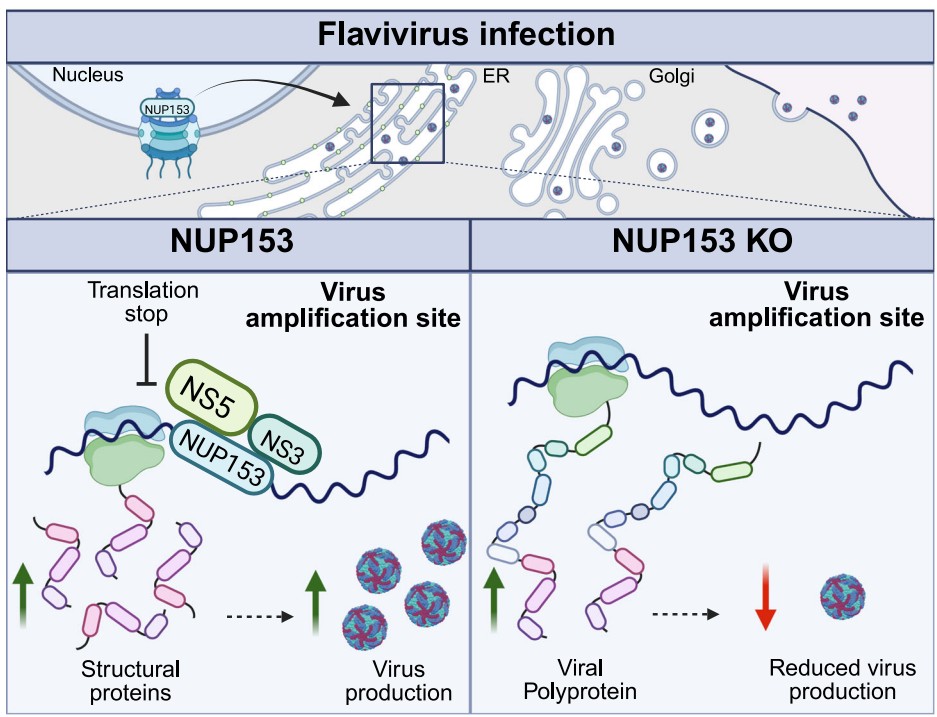

**Fig. 9 | Schematic diagram of NUP153's function during orthoflavivirus infection.** NUP153 is found at the virus amplification site during orthoflavivirus infection, where it interacts with viral RNA and viral proteins NS3 and NS5 to modulate the expression of structural proteins leading to increased virion production and release.

characterized[40]. Previous studies of TBEV and other orthoflaviviruses have shown that G4-stabilizing ligands lead to reduced infection, highlighting the importance of G4 unfolding mechanisms during infection[39,53]. Furthermore, several RNA-binding proteins that bind G-rich RNA are involved in G4 unfolding in eukaryotic cells. Since NUP153 binds to a site in the viral RNA that contains several conserved guanines, it might be such a protein. However, NUP153 did not bind efficiently to G4-containing RNA in vitro, which suggests that other proteins are needed to unfold G4s to allow NUP153 binding. In fact, based on the protein interactome data presented in the current study, approximately 50% of proteins pulled down with NUP153 are predicted to bind G4s (QUADRatlas)[40,54]. These protein interactions were not changed during infection (Supplementary Data 1), and they could therefore be important for the interaction between NUP153 and G-rich RNA[40,55]. Viral proteins are also able to unfold G4s, e.g., HCV NS3, an equivalent of orthoflavivirus NS3 that we showed here interacts with NUP153[56,57]. Hence, both cellular and viral components could be involved in the unfolding of viral G4 structures to allow NUP153 to bind and promote effective virus replication. However, further studies are required to understand this interplay.

The data presented herein indicate that NUP153 is an important factor involved in regulating the ratio of the different orthoflavivirus proteins. Namely, it binds viral RNA to affect the amount of specific viral proteins to promote infection spread. Currently, mechanisms regulating the translation of the single polyprotein encoded by the orthoflavivirus genome during viral replication are not well understood. Generally, viruses use different strategies to maximize the genome-coding capacity, such as programmed ribosomal frameshift, reinitiation, stop codon readthrough, leaky scanning, and translational bypass, as reviewed by Stern-Ginossar et al.[58]. It has been suggested that WNV employs a ribosomal frame shift, but the sequences required for this mechanism are poorly conserved among orthoflaviviruses[59]. It has been also shown that orthoflavivirus NS5 regulates the switch between translation and replication of viral RNA[43]. However, overall, regulation of protein translation is poorly understood, and it is generally thought that the orthoflavivirus polyprotein yields equal

amounts of the 10 constituent proteins, upon proteolytic cleavage. This contrasts with what has been reported for viruses that regulate protein amounts at the transcriptional level; for example, the measles virus produces much more nucleocapsid protein than L protein[60]. In the current study, we observed that NUP153 depletion results in a reduced expression of the E protein and an increased expression of several nonstructural proteins during orthoflavivirus infection, indicating its involvement in regulating protein ratios (Fig. 8). This regulation is likely mediated by binding of NUP153 to viral RNA between structural and nonstructural proteins (Fig. 5C) and to the ribosome (Fig. 3B). Interestingly, NUP153 has been reported to interact with polysomes during ZIKV and DENV infection of Huh7 cells[61]. This interaction seemed to be transient as polysomes containing NUP153 only appeared 60 hpi for DENV and 24 hpi for ZIKV. We could also show in an in vitro assay that addition of NUP153 RBD resulted in a decrease of luciferase activity downstream of the peak1 and production of a lower molecular weight protein that we believe to be the C-prM-E polyprotein, in contrast, an increased luciferase production was detected in the mutant construct when the upstream peak1 was mutated but not the downstream peak1 (Fig. 8A, D). Although our data suggest NUP153 might be playing a regulatory role in viral protein ratio, further studies using translation specific technologies, such as ribosomal profiling in virus-infected or replicon expressing cells will be crucial to understand if and how NUP153 regulates polyprotein translation elongation in an in vivo context. Generally, more studies on proteins found in orthoflavivirus polysomes are also required to fully understand how cell translation and gene expression in general are regulated during infection.

Overall, we propose a previously undescribed role of NUP153 as a proviral factor in orthoflavivirus infection, specifically, modulation of viral protein ratios to support the spread of infection (Fig. 9). The proviral role and NUP153 involvement is conserved in both mosquito- and tick-borne orthoflaviviruses, which makes this mechanism an interesting potential target for broad spectrum antiviral development. This study underscores the importance of investigating the role of NUPs in infection and the regulation of viral protein translation.

## Methods

### Cell culture and reagents

A549 and HEK293T cells are both good models for TBEV infection, A549 cells are better for immunofluorescence, and HEK293T cells are better for transfection. They were cultured in Dulbecco's modified Eagle's medium (DMEM; Sigma) supplemented with 5% fetal bovine serum (FBS; Gibco), 20 units/ml penicillin, and 20 μg/ml streptomycin (GE Healthcare) at 37 °C in 5% $CO_2$. A549 MAVS$^{-/-}$ cells[16] were cultured in DMEM supplemented with FBS, penicillin and streptomycin but also supplemented with 2 μg/ml puromycin (Gibco) at 37 °C in 5% $CO_2$. VeroB4 cells were cultured in Hyclone Medium 199 with Earle´s Balanced Salts and L-Glutamine (Medium199/EBSS; GE Healthcare) supplemented with 5% FBS, 20 units/ml penicillin, and 20 μg/ml streptomycin at 37 °C in 5% $CO_2$. DLD-1 and NUP153(NG)$^{AID}$ cells were a kind gift from M. Dasso (Division of Molecular and Cellular Biology, National Institute of Child Health and Human Development, National Institutes of Health, Bethesda, MD, 20892, USA) and were cultured in DMEM + GlutaMAX (Gibco) media supplemented with 10% FBS, 20 units/ml penicillin, and 20 μg/ml streptomycin at 37 °C in 5% $CO_2$[18]. DLD-1 parental cells were engineered to express Neon green and AID tagged NUP153 (NUP153 (NG)$^{AID}$), the DLD-1 control cells did not express AID tag[18]. Auxin (1 mM) was added to degrade NUP153 in NUP153(NG)$^{AID}$ cells. Cells were transfected using GeneJuice transfection reagent (Novagen) according to the manufacturer's instructions. Brefeldin A (BFA) (Sigma) was used at a concentration of 5 μg/mL.

### Viruses

TBEV-Torö or TBEV-Neudoerfl was produced in VeroB4 cells and harvested 72 hpi[50]. LGTV was produced in A549 MAVS$^{-/-}$ cells and harvested 72 hpi. Japanese encephalitis virus (JEV) (Nakayama strain), West Nile virus (WNV) (WNV_0304h_ISR00), yellow fever virus (YFV) (Asibi), and dengue virus (DENV) (serotype-2; PNG/New Guinea C) were kind gifts from S. Vene (the Public Health Agency of Sweden) and were grown in VeroB4 cells. ZIKV (MR766) was a kind gift from G. Dobler (Bundeswehr Institute of Microbiology, Munich, Germany) and was grown in VeroB4 cells. Viruses where titrated by focus forming unit (FFU) assay on VeroB4, and FFU was determined 48 hpi by counting E-stained (anti-TBEV E 1783.3 or anti-panFlaviE) foci[50]. Titration was performed on the different cell lines to ensure a correct MOI dose.

### Plasmids

Mammalian expression plasmids (backbone pI.18) encoding C-terminally 3×FLAG-tagged TBEV-Hypr NS3; NS2B3 and NS5[50]. eGFP-NUP153, eGFP-NUP153$_{1-650}$ and eGFP-NUP153$_{880-1475}$ were a gift from B. Fahrenkrog (Addgene plasmid # 64,268; http://n2t.net/addgene:64268; RRID: Addgene_64,268; Addgene plasmid # 64,318; http://n2t.net/addgene:64318; RRID: Addgene_64,318; Addgene plasmid # 64,317; http://n2t.net/addgene:64317; RRID: Addgene_64,317). eGFP-NUP153$_{1-250}$, eGFP-NUP153$_{1-650-RBD}$ and eGFP-NUP153$_{RBD}$ were cloned using standard PCR cloning methods. Plasmids for protein expression of His-eGFP and His-eGFP-NUP153$_{RBD}$ were cloned using standard PCR cloning methods into the pQE-30Xa (Qiagen) vector. Fragments used to clone plasmids for reverse transcription were ordered from Genewiz and cloned by standard methods into pcDNA3.1. WT construct used in the in vitro translation assay was cloned by standard cloning methods. Fragment to mutate peak 1 in the Mut-28G construct was ordered from Eurofins Genomics and cloned by standard cloning methods into the WT construct used for the in vitro translation assay. All plasmids were sequenced to verify their correctness.

### Immunofluorescence

Cells were grown in 8-well chamber slides (Sarstedt), transfected and/or infected and incubated for the given amount of time. Samples for immunofluorescence were infected with a MOI 10 to ensure infection in all cells. Samples in which nucleoporin antibodies were used were permeabilized with 0.02% Triton X-100 in PBS for 30 s before fixation. They were fixed with 4% formaldehyde for 30 min and incubated in permeabilization buffer (PBS containing 2% goat serum and 0.3% Triton-X) containing primary or secondary antibodies for 1 h each at room temperature (Supplementary Table 1). Samples were washed with PBS-Tween in between the incubations and also after the secondary antibody before being mounted using mounting media. Samples were imaged and analyzed using a Leica SP8 Laser Scanning Confocal Microscope with a 63× oil objective (Leica) and Leica Application Suit X software (LAS X, Leica). For SIM cells were imaged with Zeiss Elyra 7 microscope with Lattice-SIM2 with a Plan-Apochromat 63×/1.4 oil objective (Zeiss).

### Fluorescent in situ hybridization (FISH)

A549 cells were grown in 8-well chamber slides (Sarstedt) and infected with TBEV MOI 10 for 24 h. Cells were fixed with 4% formaldehyde for 30 min and washed with PBS. They were then incubated with 70% ethanol overnight at 4 °C. Cells were then washed with wash buffer (2× saline-sodium citrate, 10% formamide) and incubated overnight with probes (poly-T 1:2000) and primary antibody (NS3 1:500) in hybridization buffer (2× saline-sodium citrate, 10% dextran sulfate, 10% formamide) at 37 °C. Cells were washed once and incubated 1 h in wash buffer containing secondary antibody and DAPI. Cells were washed once in wash buffer and mounted using mounting media. Samples were imaged and analyzed using a Leica SP8 Laser Scanning Confocal Microscope with a 63× oil objective (Leica) and Leica Application Suit X software (LAS X, Leica).

### Image analysis and quantification

Images obtained from the confocal microscope were processed using ImageJ[62], and quantification were then done using CellProfiler[63]. The DAPI channel was used to identify the nuclei as primary objects, while GAPDH or expansion of the DAPI channel was used to identify the whole cell as secondary object. These two objects were then used to identify the cytoplasmic fraction as a tertiary object. "MeasureObjectIntensity" and "MeasureColocalization" were then used to analyze the stained proteins in the different fractions. Microsoft Excel was then used to process tables for the selection of infected or transfected cells. Final plots were then generated using GraphPad Prism.

SIM images were processed and quantified using Imaris software (version 10.2) with the ImarisColoc package used to determine the Manders' coefficient in entire images.

### RT-qPCR

Samples for RNA extraction were infected with MOI 10 to ensure infection in all cells and also obtain comparable results to the data obtained from immunofluorescence assays. In experiments performed on NUP153(NG)$^{AID}$ cells an MOI of 0.5 was used to be able to compare results with other infectious assay read outs. RNA was isolated using the NucleoSpin RNA Plus (Macherey Nagel) kit according to manufacturer's instructions. cDNA was synthesized using high-capacity cDNA Reverse Transcription kit (Thermo Fisher). qPCRBIO SyGreen mix Hi-ROX (PCR Biosystems) was used to detect *IFNB1* RNA (QT00225372, Qiagen) and *NUP153* RNA (primers found in Supplementary Table 2). qPCRBIO probe mix Hi-ROX (PCR Biosystems) was used to detect LGTV and TBEV RNA (primers found in Supplementary Table 2). Samples were normalized to *actin* (QT01680476, Qiagen) that was used as reference gene. RT-qPCR experiments were run on a StepOnePlus real-time PCR system (Applied Biosystems).

### Plate reader immunofluorescence assay

Twenty-five thousand cells were seeded into CELLSTAR® 96-well (Greiner) plates and infected with different viruses using approximately MOI 0.5, titrated on the specific cell lines used, to reach

approximately 50% infection at the end of the experiment. Cells were then fixed for 30 min using 4% formaldehyde and washed once with PBS. They were then permeabilized for 10 min with permeabilization buffer (0.5% Triton-X, 20 mM glycine in PBS) at room temperature. Cells were then incubated in primary antibody in dilution buffer (10% FBS, 0.05% Tween-80 in PBS) for 1 h. They were then washed three times with PBS-Tween and incubated 1 h with secondary antibody diluted in dilution buffer. Cells were washed three times with PBS and imaged and analyzed with Cytation 5 and BioTek Gen5 software (Agilent).

### Affinity purification-mass spectrometry

HEK293T cells were seeded in T-175 flasks and transfected with 10 μg eGFP-NUP153 or eGFP. They were infected with TBEV-Neudörfl (MOI 5) for 1 h at 37 °C and 5% $CO_2$. Then the inoculum was removed and replaced with fresh medium. After 24 h of infection cells were washed in PBS and then lysed in cold lysis buffer (10 mM Tris-HCl, pH 7.5, 150 mM NaCl, 1% NP-40 substitute (Sigma 74385), and 2x Protease inhibitor (Roche, cOmplete, Mini, EDTA-free, 4693159001) for 1 h. Lysates were then centrifuged at $16,000 \times g$, 4 °C for 15 min. Supernatants were transferred to new tubes and were flash frozen and stored at −80 °C.

Total protein concentration in HEK293T cell lysates expressing eGFP or eGFP-NUP153, infected with TBEV-Neudörfl or mock infected was quantified using the DC assay (Bio-Rad). Three biological replicates per condition were prepared.

The lysates were diluted to 0.4 mg/ml protein concentration using dilution buffer (10 mM Tris-HCl, pH 7.5, 150 mM NaCl, 1x Protease inhibitors), and 1250 μl of this (0.5 mg protein) was used for each sample. The diluted cell lysate was incubated with 25 μl of GFP-trap® Dynabeads™ (Chromotek) at 4 °C for 1 h on a rotator. The beads were then washed three times with wash buffer (10 mM Tris-HCl, pH 7.5, 150 mM NaCl) and the binding proteins were eluted using 50 μl acidic elution buffer (200 mM glycine-HCl, pH 2.5) while vigorously pipetting. The eluate was neutralized with 10 μl neutralization buffer (1 M ammonium bicarbonate), then reduced with DTT and alkylated using IAA. The proteins were digested using trypsin on 37 °C overnight. The digested samples were then acidified using 6.5 μl acidifying solution (83.3% acetonitrile (AcN), 16.7% TFA) to pH <3. The peptides were desalted using in-house-made STageTips[64,65]. Briefly, the STageTip was constructed using 2 layers of C18 membrane (3 M Empore) packed into a 200 μl pipette tip. The membrane was activated with methanol, then washed with 80% AcN, 0.1% formic acid once and twice with 0.1% formic acid. The acidified samples were loaded onto the tips, washed with 0.1% formic acid and eluted with 80% AcN, 0.1% formic acid. The eluate was then vacuum-dried and stored at −80 °C until use.

The instrumental setup consisted of an Easy-nLC 1000 nanoLC (Thermo Fisher Scientific) equipped with an Acclaim PepMap 100 pre-column (Thermo, 75 μm × 2 cm, 3 μm, 100 Å) and an EASYSpray Pep-Map RSLC C18 analytical column (Thermo Fisher Scientific, EASYspray, 75 μm × 15 cm, 2 μm, 100 Å), while the mass spectrometer (MS) used was a QExactive Plus instrument (Thermo Fisher Scientific). The EASYSpray source was operated on 40 °C, and the spray voltage was set to 1.9 kV. The liquid chromatography gradient method went from 4 to 76% acetonitrile in 79 min with both mobile phases containing 0.1% formic acid. The MS was used in the positive ion mode, the full scan having a resolution of 140,000 (400–1700 m/z) and an automatic gain control (AGC) target of $3 \times 10^6$, while the resolution and AGC target was 17,500 and $1 \times 10^5$ for MS/MS, respectively. The top 10 most abundant ions were fragmented and measured in MS/MS using data-dependent acquisition with dynamic exclusion of 30 s enabled.

The generated raw files were searched using MaxQuant[66] (2.0.1.0) against the human FASTA file (Uniprot, Homo sapiens, reviewed, 20,360 entries, downloaded on 2022-02-21.) supplemented with TBEV (U27495.1, protein sequences downloaded from NCBI (AAA86870.1) on 2022-04-06.) and LGTV (strain TP21, protein sequences

downloaded from NCBI (NC_003690.1) on 2022.04.06) protein sequences. Trypsin/P was used as the digestion enzyme with maximum 2 missed cleavages allowed. Carbamidomethylation of cysteines was selected as fixed modification, while methionine oxidation and N-terminal acetylation was used as variable modifications. Label-free quantification (MaxLFQ algorithm[67]) was selected with a minimum ratio count of 2. Peptide mass tolerances were 20 and 4.5 ppm for the first and main search, respectively.

PSM and protein FDR was set to 0.01. The minimum number of detected peptides required was 2, and the minimum number of unique peptides was 1 for identification.

As viral proteins are often short and have many basic residues, they are more challenging to identify in regular searches. For this reason, specific searches were carried out. First with MaxQuant (2.0.1.0) using only the specific viral sequences as the FASTA file and lowering the minimum peptide length for unspecific search to 4 amino acids (7 and 8 in the original search, respectively), and only 1 peptide and 1 unique peptide was required. These searches are, however, biased and can result in many false positives. The search was repeated in Proteome Discoverer (1.4.0.288) (Thermo Fisher Scientific) using Sequest HT and Target Decoy PSM validator. The corresponding settings to an Orbitrap instrument were selected, and the same modifications used as above. The target FDR was 0.01 (strict) and 0.05 (relaxed). For protein identification, the high-confidence peptide fragment spectra were observed so that the sequence identity could be confirmed. The peptides were also compared to the human proteome using BLAST to validate that they are not human peptide hits.

### Statistical analysis (AP-MS)

The generated LFQ intensity data from the MaxQuant output file "proteinGroups.txt" was first used to identify proteins found uniquely in NUP153 either TBEV, and mock-infected, and then further processed for statistical testing employing Perseus (2.0.3.0)[68], selecting the LFQ intensity values as main. Proteins found in more than 1 out of 6 GFP samples were removed from the analysis. Potential contaminants, proteins only identified by site or reverse hits were removed, and the LFQ intensities transformed to a log2(x) base. The two sample groups were defined, then the proteins were filtered based on valid intensity values in the samples (minimum 2 valid values in both groups). Missing values were imputed using values from normal distribution (width 0.3, down shift 1.8, total matrix). To determine the significantly enriched proteins a two-sided t-test was used (permutation-based FDR:0.05, S0:0.1, 250 permutations).

### Protein synthesis measurement using OPP

Cells were induced with auxin 2 h prior to infection with TBEV MOI 10. A high MOI was selected to ensure all cells were infected. Global translation was measured using the OPP assay (OPP, Thermo Fisher Scientific) following the manufacturer's instructions. In brief, cells were treated with 5 μM OPP for 30 min at 37 °C in 5% $CO_2$. The cells were then fixed for 30 min using 4% formaldehyde and permeabilized in 0.5% Triton X-100 for 15 min. OPP was then fluorescently labeled 30 min by Click-iT Plus OPP reaction cocktail with Alexa Fluor 647 picolyl azide. Cells were then analyzed by Cytation5 (Agilent) by measuring the mean fluorescent intensity.

### Trypan blue viability assay

Cells were induced with auxin 2 h prior to infection with TBEV (MOI 10). Following treatment cells were detached using 0.25% trypsin. Cells were mixed in a 1:1 ratio with Trypan Blue stain 0.4% (Invitrogen) and measured with the countess II automated cell counter (Invitrogen).

### Co-immunoprecipitation

HEK293T cells were seeded in T-75 (Sarstedt) and transfected with 6 μg of the plasmids of interest and infected with TBEV or LGTV (MOI 1) for

48 h. Cells were washed once in PBS and lysed using lysis buffer (50 mM Tris [pH 8.0], 150 mM NaCl, 1% Triton X-100 and protease inhibitor [1 tablet per 5 mL]). For RNase treatment lysis buffer was supplemented with 20 µg/mL RNase A (Thermo Fisher Scientific). Cell lysates were centrifuged at $15,000 \times g$, 4 °C for 10 min, and the supernatants were precleared by the addition of 25 µL control beads (Chromotek) and rotated for 2 h at 4 °C. Beads were removed by centrifugation, and 60 µl of cell lysate was saved for the input control. GFP precipitation was performed on the remaining precleared sample using 25 µl GFP-trap beads (Chromotek) and rotated for 1 h at 4 °C. GFP-trap beads were washed four times in lysis buffer, and proteins were eluted by incubation at 95 °C 2× Laemmli sample buffer prior to Western blotting.

## SDS-PAGE and Western blot

Cell lysate or co-immunoprecipitation samples were separated by SDS-PAGE using precast bolt 4%-to-12% Bis-Tris gels (Invitrogen). PageBlue protein staining solution (Thermo Scientific) was used according to manufacturer's instructions to visualize proteins on SDS-PAGE. For Western blots, proteins were transferred onto an Immobilon-P polyvinylidene difluoride (PVDF) membrane (Millipore), followed by incubation in blocking buffer (0.02% ECL Prime Blocking Reagent [Amersham] diluted in 0.05% PBS-Tween). The membrane was first incubated with primary antibody overnight at 4 °C, washed three times with PBS-Tween, and then incubated with fluorescent secondary antibody for 1 h at room temperature. After washing steps, band detection was performed using the Amersham Imager 680 (GE Life Sciences).

## Cross-link immunoprecipitation (CLIP)

HEK293T cells were seeded in 6-well plates and let to settle overnight. The cells were then transfected with 1.5 µg plasmid of the proteins of interest for 24 h and infected with LGTV (MOI 1) for 48 h. Culture media was removed and cells were irradiated at 250 mJ/cm² (254-nm wavelength) using a Spectro Linker XL-100 UV crosslinker (Spectronics Corporation) to cross-link the proteins with the RNA. They were then detached using ice-cold PBS and transferred to 1.5 mL tubes. Samples were centrifuged at $250 \times g$ for 10 min. Supernatant was removed and cells were resuspended and lysed in 400 µL lysis buffer (50 mM Tris-HCl [pH 8.0], 150 mM NaCl, 1.5% Triton X-100, protease inhibitor cocktail [Roche], 5 U/µl RNase inhibitor SUPERase-In [Thermo Fisher Scientific]) for 30 min on ice. Samples were then centrifuged at $16,000 \times g$ at 4 °C for 15 min and supernatants were transferred to new Eppendorf tubes. Protein A agarose beads (Merck) (50 µl/sample) or control beads (Chromotek) (25 µl/sample) were washed three times in lysis buffer or dilution buffer (10 mM Tris [pH 7.5], 150 mM NaCl, 0.5 mM EDTA diluted in water), added to supernatants and incubated for 2 h to preclear the lysates. Samples were then centrifuged at $1000 \times g$ and supernatants were transferred to new tubes. Fifty microliters of cell lysate was saved as input control. The precleared lysates were then incubated either overnight with antibody (anti-eGFP) or for 1 h with GFP-trap beads (Chromotek). After overnight incubation samples were then incubated with Protein A agarose beads for 1 h at 4 °C. Following incubation with either Protein A agarose beads or GFP-trap beads, beads were washed four times with wash buffer (50 mM Tris-HCl [pH 8.0], 150 mM NaCl, 1% Triton X-100). The RNA was then eluted from the beads using QIAzol (Qiagen) according to manufacture's instructions and purified and eluted in 40 µl RNA-free water using the NucleoSpin RNA kit (Macherey-Nagel) according to manufacture's instructions. cDNA (High-Capacity cDNA Reverse Transcription kit; Thermo Fisher Scientific) was synthetized from the RNA according to manufactures instructions. To quantify the viral RNA, qPCR was performed on the StepOnePlus instrument using the KAPA probe MM (Techtum) and the LGTV NS3 probes and primers[69]. Ct values of IP samples were normalized against Ct values of the respective input collected prior to immunoprecipitation. Furthermore,

values were further normalized to the control bead immunoprecipitation of each transfected construct.

## Protein purification

Twenty milliliters of LB broth (10 g/L tryptone, 5 g/L yeast extract and 10 g/L NaCl) containing 100 µg/ml ampicillin and 25 µg/ml kanamycin were inoculated with either bacteria for His-eGFP or His-eGFP-NUP153_RBD production and grown overnight at 37 °C using a bacterial shaker. One liter of LB broth containing 100 µg/ml ampicillin and 25 µg/ml kanamycin was inoculated with the overnight culture and grown at 37 °C on a bacteria shaker until an $OD_{600}$ of 0.6 was reached. Expression of protein was induced for 5 h with 1 mM IPTG. Cells were harvested by centrifugation at $4000 \times g$ for 20 min. Pellet was resuspended in lysis buffer (50 mM $NaH_2PO_4$, 300 mM NaCl and 10 mM Imidazole [pH 8]) supplemented with 1 mg/mL of lysozyme for 30 min on ice. Lysate was sonicated with a sonicator equipped with a microtip using 10 s busts at 200–300 W with a 10 s cooling period between each burst. 10 µg/mL RNase A and 5 µg DNase was added to the lysate and incubated for 15 min on ice. Lysate was centrifuged at $10,000 \times g$ for 30 min at 4 °C to clear lysate. Fifty percent Ni-NTA (Qiagen) slurry was added to cleared lysate and mix gently by shaking on a rotary shaker at 4 °C for 60 min. Lysate-Ni-NTA mixture was loaded into a polypropylene column (Qiagen) and liquid was allowed to flow through. Ni-NTA was washed twice with wash buffer (50 mM $NaH_2PO_4$, 300 mM NaCl and 20 mM Imidazole [pH 8]) and protein was eluted using elution buffer (50 mM $NaH_2PO_4$, 300 mM NaCl and 250 mM Imidazole [pH 8]) in 0.5 mL fractions. Fractions were analyzed by SDS-PAGE. Fractions containing protein were pulled and a buffer exchange to PBS was performed using the Pur-A-Lyzer Maxi 3500 (Sigma). Following this, protein was concentrated using Amicon Ultra -4 10k columns (Millipore) according to manufacturer's instructions.

The His and eGFP tags were added to the NUP153_RBD to obtain an overall more neutral isoelectric point of the protein, as the isoelectric point of the NUP153_RBD is ~12 making very positively charged which can be problematic for certain assays.

## In vitro transcription

Reverse transcription was performed according to manufacturer's instructions using the MEGAscript T7 transcription kit or MEGAscript SP6 transcription kit (Invitrogen). Following the reaction DNA was degraded with 1 µl/sample TURBO DNase (Invitrogen) and RNA was precipitated using Lithium chloride. RNA was dissolved in RNase-free water and size fragments were verified by gel electrophoresis. To label the RNA, UTP was replaced by Aminoallyl-UTP-Cy3 (Jena Bioscience) in in vitro transcription reaction.

For in vitro translation assays, in vitro transcription was performed using the mMESSAGE mMACHINE SP6 transcription kit (Thermo Fisher), to produce capped RNA.

## RNA electrophoretic mobility shift assay (RNA EMSA)

His-eGFP-NUP153-RBD or His-eGFP protein in different concentrations was mixed with 80 ng of Cy3 tagged viral RNA fragments in 10 µL RNA binding buffer (25 mM Tris pH [7.6], 100 mM KCl, 3 mM MgCl2, 0.01% Tween20, 1 mg/mL BSA and 1 mM dithiothreitol [DTT], dissolved in water) for 30 min at room temperature. Six microliters of 50% glycerol was added to each sample, loaded on a 0.8% Agarose gel and ran for 30 min at 100 V. Gels where imaged using Amersham Imager 680 (GE Life sciences). The ImageJ[62] software was used to quantify the amount of bound and unbound RNA. The data was then plotted using GraphPad Prism and the non linear fit one site specific binding function was used to approximate binding curves.

## Cross-link immunoprecipitation sequencing (CLIP-seq)

HEK293T cells were seeded in TC100 dishes (Sarstedt) and transfected with 5 µg eGFP-NUP153_1-650 plasmid. Cells were then infected with

TBEV (MOI 1) for 48 h. Culture media was removed, and cells were irradiated at 250 mJ/cm$^2$ (254-nm wavelength) to cross-link the proteins with the RNA. They were then detached using ice-cold PBS and transferred to 1.5 mL tubes. Samples were centrifuged at 250 × $g$ for 10 min. Supernatant was removed, and cells were resuspended and lysed in 500 μL RIPA buffer (50 mM Tris-HCl [pH 7.6], 150 mM NaCl, 1.5% Triton X-100, 0.5% Sodium deoxycholate and 0.1% SDS supplemented with protease inhibitor cocktail [Roche], 5 U/μl RNase inhibitor SUPERase-In [Thermo Fisher Scientific]) for 1 h at 4 °C. Samples were then centrifuged at 16,000 × g at 4 °C for 15 min and supernatants were transferred to new tubes. Control beads (Chromotek) (25 μl/sample) were washed three times in RIPA buffer and incubated for 2 h with sample at 4 °C on a rotor to pre-clear the lysates. Samples were then centrifuged at 1000 × $g$, and supernatants were transferred to new tubes. The precleared lysates were then incubated 1 h with GFP-trap beads (Chromotek) at 4 °C on a rotor. Following this RNase A (Thermo Fisher Scientific) in a concentration of 1:2500 was added to the samples and incubated for 15 min at room temperature on a rotor. GFP-trap beads were then washed four times with RIPA buffer. The RNA was then eluted from the beads using QIAzol (Qiagen) and isopropanol RNA precipitation according to manufacture's instructions. RNA fragment size and concentration were checked using the RNA 600 Pico kit (Agilent) and measured using the 2100 Bioanalyzer (Agilent).

cDNA library was generated with the NEBNext small RNA sequencing kit (New England Biolabs) according to manufacturer's instructions and purified using the DNA clean & concentrator™-5 (Zymo). Samples were then pooled, and Sera-Mag Speedbeads (Cytiva) were used at a concentration of 3× to perform one-sided purification to remove longer sequences. DNA fragment size and concentration was then verified and measured by Bioanalyzer High sensitivity DNA kit (Agilent) and Qubit dsDNA HS assay kit (Invitrogen). The resulting cDNA libraries were sequenced using the Illumina MiSeq platform.

Prior to analysis, adapters and indexes were removed using cutadapt[70] and the quality of reads was checked using FastQC. To be able to map the sequences that are unique to the virus genome, the human genome and the TBEV-Torö genome (DQ401140.3) were first concatenated using bowtie2[71]. Sequences were then aligned to the concatenated genome using bowtie, allowing for up to 3 mismatches and a maximum of 10 identical reads with the best stratum[72]. Samtools[73] was then used to sort and index the reads, and bedtools was then used to convert the files from BAM files to BED files. The galaxy platform was then used to identify significant peaks using PEAKachu[74]. Finally, Gviz[75] was used to plot the data.

## Multiple sequence alignment

Multiple sequence alignment was performed using AliView[76]. The sequences used for alignment are TBEV (DQ401140.3), LGTV (EU790644.1), POWV (NC_003687.1), JEV (NC_001437.1), WNV (NC_009942.1), YFV (NC_002031.1), DENV2 (NC_001474.2:97-10272), and ZIKV (NC012532.1).

## In vitro binding of RNA fragments to NUP153

HEK293T cells were seeded in T-75 flasks and transfected with 5 μg of eGFP-NUP153$_{1-650}$ plasmid using GeneJuice according to manufacturer's instructions. Twenty-four hours after transfection, cells were lysed with RIPA buffer supplemented with RNase A in a concentration of 1:500. At this concentration of RNase A no RNA could be detected using the RNA 600 Pico kit (Agilent) and measured using the 2100 Bioanalyzer (Agilent). Samples were then pre-cleared using control beads, and proteins were purified using GFP-trap beads. Five micrograms in vitro transcribed RNA was then added in 200 μl RNA-binding buffer (5 mM HEPES [pH 7.2], 3 mM MgCl$_2$, 5% Glycerol, 1 mM DTT and 5 mM ATP in milli-Q water, supplemented with protease and 5 U/μl RNase inhibitor SUPERase-In) with or without 100 mM KCl and cross linked with 400,000 μJ/cm$^2$ energy at 254 nM UV on ice. Samples were

then incubated overnight at 4 °C on rotor and cross-linked again with 400,000 μJ/cm$^2$ energy at 254 nM UV on ice. Samples were then washed four times in RIPA buffer, and RNA was isolated and precipitated with QIAzol and isopropanol according to manufacturer's instructions. RNA concentration was then determined using Qubit RNA HS assay kit (Invitrogen) according to manufacturer's instructions.

## Circular dichroism spectroscopy (CD)

CD experiments were carried out in Jasco-J1700 CD spectrophotometer, having built-in Peltier cell holding carousel and temperature controller. The wild-type and mutant RNA sequences were diluted to 4 μM in the buffer containing 10 mM Tris (pH 7.0), supplemented with or without 100 mM KCl and annealed by heating at 70 °C for 5 min, followed by slowly cooling down to room temperature for 24 h. Samples were scanned within 320–210 nm wavelengths at 25 °C with a scanning speed of 100 nm/min. Data points were acquired at 0.5 nm interval and averaged over 3 accumulations. For each measurement, we used quartz cuvettes of 1 mm pathlength (Hellma) having a reaction volume of 200 μL. The digital integration time and the bandwidths were 2 s and 1 nm, respectively. Ellipticity values were converted to molar ellipticity or molar circular dichroism (MCD) using the following formula and had been corrected for buffer contributions. $MCD = \frac{\theta}{32980 \times c \times l}$ $\theta$ is the ellipicity value in mdeg; while $c$ and $l$ denote the concentration of RNA (in g/L) and path-length of cuvette (in cm), respectively. To understand the effect of KCl on G4 stabilization, we conducted temperature-induced CD melting experiments in these samples using the parameters mentioned above. For thermal scans, we heated the samples from 25 to 95 °C, having temperature gradient of 5 °C/min and a delay time of 300 s. The MCD values of positive maxima were normalized to 25 °C (maximum folded G4) and 95 °C (maximum unfolded G4) using the following formula to calculate the melting temperatures ($T_m$):

Fraction of folded G4 at any temperature = $\frac{(\theta_t - \theta_U)}{(\theta_F - \theta_U)}$ where $\theta_t$ denotes MCD at any temperature, $\theta_U$ and $\theta_F$ represent the MCDs at 95 and 25 °C, respectively.

## Mass spectrometry on NUP153(NG)$^{AID}$ cells infected with WNV

NUP153(NG)$^{AID}$ were seeded in 6-well plates and let to settle overnight. They were induced with 1 mM auxin 2 h prior to infection with WNV (MOI 0.5). At 48 hpi cells were lysed using RIPA buffer (50 mM Tris-HCl [pH 8], 150 mM NaCl, 1% NP-40, 0.5% Sodium deoxycholate and 1% SDS, supplemented with protease inhibitor) for 1 h at 4 °C. Proteins were subsequently diluted with Ammonia-Bicarbonate-Buffer (100 mM NH4HCO3, pH 8, Thermo Fisher Cat#: 393212500), reduced with 10 mM DTT (Merck/Millipore, Cat#:1114740001) and alkylated with 55 mM iodoacetamide (Thermo Fisher, Cat#: 122270050). Samples were precipitated using acetonitrile (Merck/Millipore, Cat#: 1000291000) and digested with trypsin (Pierce Trypsin Protease MS Grade, Cat#: 90305) at 37 °C o/n. Tryptic peptides were next acidified with 1% Formic Acid (Carl Roth, Cat#: 1EHK.1), desalted via reverse phased C18 Tips (Fisher Scientific, Cat#: 10615555), and separated via nano high-performance liquid chromatography (Vanquish Neo, Thermo Scientific, Waltham, MA, USA). Subsequently, a nano electrospray ion source interfaced with an Orbitrap Tribrid Mass Spectrometer (Eclipse, Thermo Scientific, Waltham, MA, USA) was used to ionize peptides. Briefly, the peptides were separated using an acetonitrile gradient over a 142-min total gradient on a C18 column. Survey full-scan MS spectra were recorded in data-dependent acquisition mode at a resolution of 60,000, and MS2 spectra at a resolution of 15,000. Fragmentation was performed using higher-energy collisional dissociation with a normalized collision energy of 30. Mass spectrometric data were analyzed via the Proteome Discoverer software (version 3.1, Thermo Scientific, Waltham, MA, USA) against the Uniprot human reference proteome database (EMBL-EBI, Cambridgeshire, UK)

(last updated on 27/03/2024) using the Sequest HT search engine (The University of Washington's Proteomics Resource, Seattle, WA, USA) and against the full-length West Nile virus strain proteome. FDR was set to 1% on the peptide and protein level. Trypsin specificity allowing up to two missed cleavages, fixed carbamidomethyl modification on cysteine, and variable oxidation on methionine were set as parameters. Moreover, at least one unique peptide with a length of minimum six amino acids per protein group was demanded. Quantifications were performed using the Minora feature detector algorithms.

### In vitro translation
Prior to vitro translation, mRNA was heated at 65 °C for 3 min and let cool immediately on ice, to denature RNA. In vitro translation was performed using the Flexi Rabbit Reticulocyte Lysate System (Promega) according to manufacturers' instructions. Reactions where supplemented with 4 μM of either His-eGFP-NUP153$_{RBD}$ or His-eGFP protein. Following 90 min incubation at 30 °C, luciferase signal was immediately measured using the Luciferase Assay system (Promega) and the CLARIOstar Plus (BMG LABTECH) Luminometer. Five microliters of the reaction was then mixed with 20 μL of SDS sample buffer, heated for 10 min at 95 °C and analyzed by Western blot. Detection of translated proteins was performed with anti-TBEV_E (GTX642364) antibody using chemiluminescent detection with SuperSignal™ West Femto Maximum Sensitivity Substrate kit (Thermo Scientific) and imaged with the Amersham Imager 680 (GE Life Sciences).

### Protein stability assay
500,000 NUP153 (NG)$^{AID}$ cells were seeded into 6-well plates (Avantor, VWR). Cells were induced with auxin 2 h prior to infection with TBEV MOI 0.5. After 24 h of infection cells were treated with 50 μg/mL cycloheximide and then lysed at 0, 3, and 6 h post cycloheximide treatment in cold lysis buffer (1.5% Triton-X 100, 50 mM Tris-HCl pH 7.6, 150 mM NaCl, 0.1% SDS and 0.5% sodium deoxycholate) and 2x Protease inhibitor (Roche, cOmplete, Mini, EDTA-free, 4693159001) for 1 h. Lysates were then centrifuged at $15,000 \times g$, 4 °C for 10 min. Supernatants were transferred to new tubes and were then boiled with 1× NuPAGE™ LDS Sample Buffer (Thermo Scientific) and 2× NuPAGE™ Sample Reducing Agent (Thermo Scientific) for 10 min. Samples were then subjected to SDS-PAGE and Western blot.

### Statistical analysis
Data from immunofluorescence, RT-qPCR, Qubit, RNA EMSA, infection assays and Western blot assays were analyzed using the GraphPad Prism 10.2.3 software.

### Reporting summary
Further information on research design is available in the Nature Portfolio Reporting Summary linked to this article.

## Data availability
Source data are provided with this paper. The affinity purification mass spectrometry data generated in this study have been deposited in the PRIDE database under accession code PXD052523. The mass spectrometry data from WNV-infected cells generated in this study have been deposited to the ProteomeExchange Consortium via the PRIDE partner repository with the dataset identifier PXD067546. The CLIP-seq data generated in this study have been deposited in the Sequence Read Archive database under accession code PRJNA1137903. Source data are provided with this paper.

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

## Acknowledgements

We thank G. Dobler (Bundeswehr Institute of Microbiology, Munich, Germany) for providing stocks of LGTV TP21, TBEV Neudörfl, and ZIKV MR766 strains, and VeroB4 cells; S. Vene (Public Health Agency of Sweden) for providing stocks of JEV (Nakayama strain), WNV (WNV_0304h_ISR00), YFV (Asibi), and DENV (serotype-2; PNG/New Guinea C). DLD-1 and NUP153(NG)^AID cells were provided by M. Dasso (Division of Molecular and Cellular Biology, National Institute of Child Health and Human Development, National Institutes of Health, Bethesda, MD, USA). We thank S. Lind (Chemistry-BMC, Uppsala University, Sweden), Bettina Sarg (Protein Core Facility, Institute of Medical Biochemistry, University of Innsbruck, Austria) and the Protein Core Facility of the Medical University of Innsbruck for the help with mass spectrometry data analysis. We thank C. Patthey (Department of Radiation Sciences, Oncology, Umeå University, Umeå, Sweden) for help with the CLIP-seq experiments. Funding: Laboratory for Molecular Infection Medicine Sweden (MIMS) VR2021-06602 to A.K.Ö.; Swedish Research Council, grants 2018-05851, 2020-06224 and 2024-00390 to A.K.Ö., 2018-05851 and 2024-00390 to R. Lu., 2020-03380 to Y.I., 2021-02468 to N.S.; the Swedish Foundation for Strategic Research, grant SB16-0039 to Y.I.; Swedish Cancer Society 22 2380 Pj to N.S.; and Knut and Alice Wallenberg foundations (KAW2021-0173) to N.S., (KAW2024-0039) to A.K.Ö., R. Lu. and German Research Foundation (INST 193/90-1 FUGG; project-ID: 497694394) to G.G. We also acknowledge Umeå Center for Microbial Research (UCMR); the Biochemical Imaging Center at Umeå University (BICU), the National Microscopy Infrastructure for Microscopy Support (NMI; VR-RFI 2019-00217); and BioMolecular Characterization Umeå (BMCU) for CD spectroscopy. We also thank Elevate Science for their editorial service. Illustrations (Figs. 4A, 5A, and 8L) were created in BioRender. Överby, A (2026) https://BioRender.com/67626ap, https://BioRender.com/r30wsap, https://BioRender.com/rw0asnh, https://BioRender.com/3sto0kr.

## Author contributions

M.B.A.P., A.K.Ö. conceived the study. All authors designed the experiments. M.B.A.P. acquired and analyzed data from confocal microscopy experiments, Co-IP, CLIP assays, purified proteins, RNA-EMSA, in vitro assays, knock-down experiments, and infections, and performed bioinformatic analysis of CLIP-seq and mass spectrometry data. R. Li., processed samples for mass spectrometry and performed infection experiments. E.K. processed samples, performed and analyzed the data for the mass spectrometry analysis. W.L.Y. scientific discussions and writing. P.S. performed the CD spectroscopy experiments. I.N. processed samples for NUP153 (NG)^AID WNV-infected mass spectrometry experiment. M.B.A.P. prepared the figures. G.G., N.S., Y.I., R. Lu., and A.K.Ö. supervised the experiments. M.B.A.P., A.K.Ö. wrote the manuscript. All authors revised the manuscript.

## Funding

## Competing interests

The authors declare no competing interests.

## Additional information

**Supplementary information** The online version contains Supplementary material available at https://doi.org/10.1038/s41467-026-71449-1.

