## [Transparent Peer Review File · Nature Communications]

Proviral NUP153 binding to viral proteins and RNA regulates structural–nonstructural protein ratios in orthoflavivirus infection

Corresponding Author: Professor Anna Överby

Version 0:

Reviewer comments:

Reviewer #1

(Remarks to the Author)

Peters et al present evidence that the nuclear pore complex protein Nup153 is a proviral factor that promotes flavivirus infection. With a specific focus on tick-borne flaviviruses (TBEV and LGTV) the authors show that Nup153 is upregulated and re-localized to the cytosol during infection and that depletion of Nup153 leads to a decrease in the number of infected cells. Furthermore, they present evidence that Nup153 interacts with viral NS3 and NS5 proteins and suggest that these interactions facilitate Nup153 recruitment to the cytosol. The authors also show evidence that Nup153 interacts with viral RNA through an RNA binding domain at several G-rich regions of the viral Genome. Finally, they use light microscopy to evaluate the levels of viral structural and non-structural proteins in Nup153 depleted cells and report a change in the ratio between E and NS2B. Based on these data, the authors posit that Nup153 is involved in promoting the translation of structural proteins to facilitate virus assembly and spread.

In general, the study is well structured, using various techniques to comprehensively evaluate the role of Nup153 in flavivirus infection. My major concern with the current study is that the overall phenotype of Nup153 re-localization and the effects of Nup153 depletion on virus infection are modest. Considering that the authors also note a significant decrease in cell proliferation upon Nup153 depletion (Figure 3), it is possible or even likely that Nup153 may not have a direct role in virus infection. Rather, Nup153 may be required to maintain other cellular pathways that are needed for infection. Finally, the current data do not sufficiently support a role for Nup153 in viral translation. Substantial further experimentation is needed to support the author's conclusions.

Specific points:

1. Previous reports have shown that NUP153 is elevated in nuclear and cytosolic fractions during stress responses (PMID: 20041180). Therefore, the re-localization of Nup153 may not be virus-specific but is caused by the upregulation of Nup153 which leads to increased cytosolic protein levels that localize to the ER. Therefore, the authors need to provide sufficient controls and parallel validations to confirm the specificity of their results for virus infection. For example, ER general markers for microscopy experiments and ER resident specificity control proteins for IP experiments.

2. There is some discrepancy between the systems for Nup153 depletion in Figure 2. For the KO cells, there is a 20% reduction in Nup153 leading to a 70% reduction in infected cells while a complete depletion of Nup153 using the AID system only reduces infection by 50%. Also, while depletion experiments are performed for both LGTV and TBEV using the AID system, the KO experiments are only done for LGTV and not TBEV.

3. Throughout the manuscript, the timing and MOIs for infection experiments as well as the use of LGTV vs TBEV in specific experiments are not well justified. Considering that the authors are reporting an effect specifically at early timepoints after infection they should include detailed information on the kinetics of virus infection (RNA levels and virus titers) over the course of infection. They should also be consistent, when possible, with infection times and MOIs. Some specific examples are listed below.

a. The authors use 16hr, 24 hr or 48 hr for TBEV, 24 h or 48 h for LGTV and 48 hr for all other flaviviruses without justification. These differences make it difficult to compare results between experiments.

b. There are significant differences in MOI used for experiments with no explanation as to why. For example, the virus spread experiment in Figure 6G was done with an MOI of 0.05 for TBEV and an MOI of 5 for WNV. For this example, it is not

clear how a virus spread experiment can be performed with a starting MOI of 5 which should infect nearly all cells with the initial infection. Similar MOI discrepancies are seen throughout the manuscript and should be addressed.

c. In Fig 2 the analysis of LGTV and TBEV is inconsistent. Time of addition experiments for LGTV should also be shown as well as FFU assays for all time points. These data would be useful for identifying the specific effects of Nup153 depletion timing.

4. The authors state that Nup153 is localized to the “amplification site” and not “replication vesicles” based on images and quantification in Figure 1H-I. First, the differences between these two regions are not well defined. Second, since previous reports have shown that replication vesicles form within rough-ER sheets that also support translation (see PMID: 26073783, 28249158, 19380115) it is not clear how sites of replication and amplification can be resolved using confocal microscopy.

5. The function of Nup153 in promoting structural protein translation is not well supported by the data. The authors see only a very modest change in the levels of 2 viral proteins (E and NS2B) which could be explained through changes in cell growth or protein stability. These findings should be validated by western blot and additional structural and non-structural proteins evaluated.

6. Further to the previous point, the authors identify an interaction between Nup153 and viral RNA, but it is not clear if Nup153 binding is specific to viral RNA or just general RNA binding activity. Less than 1.4% of the reads in the CLIP-Seq were associated with viral RNA at a time point where the viral RNA should be highly abundant. If Nup153 is significantly impacting translation, the amount bound to viral RNA at this point in infection should be much higher.

7. In Figure 4, the relevance of the Nup153 RBD for infection should be validated with RBD specific deletions in the full-length Nup153. Rescue experiments should also be done to show that expression of RBD containing Nup153 variants can rescue Nup153 depletion during TBEV or LGTV infection.

Minor comments

- The CLIP sequencing experiment in Figure 4 shows several other peaks including a strong peak in the 3'UTR that is not discussed.
- The authors state that the interaction between Nup153 and C is RNA dependent (Fig3H) but there appears to be a loss of C in the input when RNase is added in all cases.
- Several of the western blots show multiple species for Nup153 without explanation or molecular mass indicators.

Reviewer #2

(Remarks to the Author)

In this study, Peters and colleagues have studied the relationship between the nuclear pore protein NUP153 and flaviviruses (mostly tick-borne viruses (TBEV and LGTV)). They show that this host factor is overexpressed in infected cells and is relocalized to the cytoplasm although the latter must be analyzed further. The authors nicely show that NUP153 binds to the viral RNA and that its protein interactome is modulated upon infection. Since NUP153 is required for cell survival, the authors took advantage of an inducible system to rapidly degrade NUP153 and assess its role during viral infection. They further propose that NUP153 regulates flavivirus life cycle at the translational level by changing the expression ratio between structural and non-structural proteins. Unfortunately, as explained below such model is not supported by the data in my opinion since no translation assays was performed. Furthermore, the requirement of NUP153 for TBEV replication remains to be confirmed.

Major comments:

1- The included data of Figure 1 were not entirely convincing regarding a relocalization of NUP153. Indeed, as shown in Fig 1D-E, NUP153 is overexpressed in LGTV and TBEV-infected A549 cells at both protein and mRNA level. Thus, the observed cytoplasmic signal in Fig 1A could simply be due to the fact that the sensitivity is higher. Actually, the nuclear signal increases as well in the same order of magnitude. To claim that there is a relocalization of NUP153 to the flavivirus replication compartment, the cytosol signal/nuclear signal ratio is expected to increase and should be determined in Fig 1A-C and 6A-B. According to the data of Fig 1C, it seems not to change much. Thus, the apparent cytoplasmic signal would be due to a global overexpression, and this would not be a relocalization process. In that case, it would only demonstrate that NUP153 colocalizes with NS3. To support their conclusion, the authors have performed sub-cellular fractionation, and the resulting quantification showed a significant increase of NUP153 in the endoplasmic reticulum (ER) fraction. However, the corresponding immunoblotting are importantly missing and must be shown. In addition, markers of the nucleus, mitochondria, ER and vesicles should be used to ensure that the fractionation process was actually efficient in separating the organelles.

Fig 1H clearly shows some colocalization between NUP153 and dsRNA. This is not consistent with the quantification shown in Fig 1I. Such colocalization was expected in my view given that NS3 mostly associates with replication complexes and that NUP153 associates with both the minus (-) and plus (+) strands of viral RNA (Fig 4B). This is not consistent. In the same line of idea, the authors conclude with Fig 4B that NUP153 preferentially associates with the (+) strand, but the data show that it also associates with the (-) strand. Such binding is not trivial given that there are many more copies of (+) strands than (-) strands in the replication compartment (which is not measured here). Considering this, Fig 4B would support the conclusion that NUP13 associates with replication complexes. In any case, the input (-) and (+) RNA levels (i.e., before IP) should be measured.

2- The potential proviral role of NUP153 needs to be confirmed. First, all the replication assays were performed by measuring the % of infected cells. This is not the best readout as it measures the capacity of the virus to spread in culture but

not directly its replication efficiency. All the results in Fig 2H, 6C and 6G should be confirmed by plaque/FFU assays and/or RT-qPCR (ideally both) which provide a better dynamic window than an imaging/FACS approach. More importantly, Fig 2I is problematic since no reduction in particle production is observed, implying that NUP153 depleted cells are still able to resume a full life cycle. This was measured at 6 hours post-auxin addition, a time point at which the phenotype of the % of infected cells is not obvious (Fig 2H). This should be performed when the treatment is optimal (e.g. addition at 2 hours post-infection). Furthermore, Fig 3F shows that NUP153 depletion reduces cell proliferation, which may be detrimental for viral replication in general, that is, not only because of the lack of NUP153 in the replication compartment. Without a clear impact of NUP153 depletion on vRNA levels and/or viral particle production (which both rely on efficient viral translation) for same number of infected cells, it cannot be concluded at that stage that NUP153 modulate flavivirus life cycle.

3- Mostly based on NUP153-dependent differences in NS3, NS2B and E protein expression levels, the authors conclude that viral RNA translation is regulated by NUP153. However, this was never addressed in this study in proper translation-based assays. Moreover, as the authors mention in the discussion, the 10 structural and nonstructural proteins are produced from the same open reading frame as a polyprotein and hence, at an equimolar ratio following processing. So, it is really hard to envision how a modulation of translation would explain the changes in NS1/E and NS2B/E protein ratios upon NUP153 depletion. Since E coding region is upstream that of NS proteins (more abundant in depleted cells), how would that work in the absence of a stop codon? Would there be an alternative translation initiation site upstream the NS1 coding region (which requires E amino-terminus to be inserted in the ER membrane). It is much more likely that viral protein expression level is modulated at the posttranslational level. Moreover, in that specific case, quantifying viral protein expression using microscopy is not appropriate since some protein population might be less accessible to the primary antibodies without allowing the monitoring of protein molecular weight/maturation. This must be done using immunoblotting. Without addressing translation per se, the model proposed in Fig 6H is not supported and really hard to conceptually comprehend, especially since it goes against the current knowledge of flaviviral polyprotein production. Using a reverse genetics system of TBEV could help characterize mutated genomes (which do not bind to NUP153) during translation and viral RNA replication in transfected cells.

Other comments:

- How is NUP153 overexpressed? Since these experiments were performed in interferon competent A549 cells, is this the consequence of the TBEV/LGTV-induced innate immune response? The authors mention MAVS knockout A549 cells in the Methods section. Is the infection-related induction of NUP153 expression also observed in these immune-deficient cells?
- Are the observed phenotypes specific to NUP153? Other NUPs should be tested for colocalization with viral proteins and potential pro/antiviral roles.
- Line 41: “, and virion assembly we define...”. Please rephrase
- Line 51: “has is required”. This should be rephrased.
- Line 52: To my knowledge, HCV does not induce convoluted membranes. Moreover, reference 15 relates to a study on HIV and not HCV.
- Line 53: “similar but different...”. This reads strange.
- Line 88: NS3 is not located in the the ER (i.e. in its lumen) but is anchored to its cytosolic side by interacting with its co-factor NS2B
- Line 160: Examples of identified proteins involved in cell cycle regulation should be indicated.
- Fig 3H: Why is there a shift in the size of C in the input upon RNase treatment?
- Line 178 : The fact that RNase treatment did not disrupt the interaction between NUP153 and NS3 or NS5 does preclude that it is indirect (e.g., mediated by other proteins). This sentence must be rephrased.
- Line 180: “flaviviruses” (typo)
- Fig 4E: The input RNA level should be measured to determine whether the -strand/+strand ratio is the same in the IP.
- Fig 5: Is there a change in the binding of NUP153 to cellular mRNAs. Such analysis of the data acquired in the CLIP would significantly strengthen the study and would be consistent with the observed change in NUP153 protein interactome.
- Line 370: This conclusion is not supported by the data as there is no experiment demonstrating the importance of the conserved G-rich region during replication. This statement must be tuned down.
- Line 402: This statement must be rephrased as the observed phenotypes on viral protein abundance could be very well due to a posttranslational regulation.

Reviewer #3

(Remarks to the Author)

The manuscript submitted by Peters, et al. is an interesting study suggesting that a nuclear pore protein, NUP153, is a proviral factor for TBEV and several other members of the Orthoflavivirus genus. The authors suggest that NUP153 is co-opted during flavivirus infection and recruited from the nucleus to the ER, where it regulates viral polyprotein translation. The data presented convincingly demonstrates that NUP153 is upregulated in response to tick-borne flavivirus infection and suggests some relocalization from the nucleus to the ER upon infection. Further, the authors present evidence showing that the RNA binding domain of NUP153 is required for relocalization and that NUP153 interacts with flavivirus RNA at G-quadruplex structures. The authors go on to suggest that NUP153 regulate translation of the flavivirus polyprotein via the interaction with viral RNA at the C-terminal portion of the E protein coding sequence, resulting in a relative increase in the translation of the upstream structural proteins, as compared to the downstream nonstructural proteins. While this is a novel and compelling model, the data presented in the manuscript do not sufficiently support these claims. Further, the addition of controls and orthogonal approaches to bolster these data are required to support many of the other conclusions.

Major Comments

1. In Figure 1A, the immunofluorescence images presented to suggest co-localization and re-localization require controls for

proper interpretation. The overall signal of NUP153 appears to increase everywhere in the cell, which supports upregulation of this protein during infection. However, the evidence that abundance is increasing at viral replication sites is not convincing and requires further experimentation, including cellular fractionation with appropriate cellular component controls. Both an ER marker and another nuclear envelope protein should be included in these experiments and in IF experiments throughout the manuscript.

2. Similarly, markers for all compartments should be included in fractionation experiments (Figure S1E). It is possible that the integrity of the nuclear envelope compromised during infection, resulting in a change in localization of nuclear proteins. Why are calnexin and GAPDH present in the pellet of all fractions in uninfected cells? This suggests the fractionation of specific compartments is not robust. Additional cellular markers, including mitochondrial and another nuclear marker should be included in this experiment. Further, the quantification of the IF data in Figure 1C suggests an increase in NUP153 in the nucleus but the quantitation of the fractionation data in Figure 1G does not reflect this. This discrepancy should be addressed.

3. Figure 1D does demonstrate an increase in NUP153 RNA and but the increase in protein is rather modest and does not appear equivalent to the increase in signal shown in Figure 1A. Further, the authors claim that this data distinguishes between increased expression and increased stability of NUP153. As NUP153 protein stability assays have not been performed here, this claim is not supported.

4. The transient knock-down of NUP153 is very inefficient, despite being performed twice and sorting on GFP positive (Cas9 positive) cells. While there is a remarkable decrease in infection in these cells, it is difficult to compare to the control (GBF1) as no knock-down data is provided here. Transfection of siRNA or shRNA targeting NUP153 would also provide transient depletion at increased efficiency. Further, this would also allow for additional infection assays, such as quantitation of FFU or PFU in the cells to further support a proviral role for NUP153 and viral life cycle assays to supplement the findings from the time of addition assays.

5. The NUP153(NG)AID system appears very robust but the infection phenotype is less pronounced here than in the transient knock-down experiment (Figure 2B) despite the complete depletion in AID system. This suggests that the phenotype in Figure 2B is not solely dependent on NUP153. The expression of NUP153 post-auxin addition should be shown for all time points. After the addition of auxin, at what point does expression of NUP153 recover? Additional viral infection assays, including the FFU assay should be included for all time points, not just the 6h timepoint at which there is no effect on infection.

6. In general, the description of experimental design, assays, normalizations and reagents used is insufficient. For several experiments, it is not clear what is being used as a negative control nor what the assays are precisely measuring. For example, in Figure 2A, data is normalized to CAS9. Is this Cas9 alone or with the non-targeting guide RNAs? Similarly, a description of the NUP153(NG)AID and control cell lines used needs a clearer description to be able to adequately interpret the results.

7. In Figure 3, the differences in NUP153 interactors in mock vs. infected cells is interesting and potentially informative for function. However, there are no experiments validating these differential interaction results.

8. In Figures 4C-E, another unrelated RNA binding protein should be used as a control. Additionally, a non-viral RNA should be quantified in 4C-4E to show specificity of interaction with viral RNA.

9. The results in Figure 5 are well-done and suggest that NUP153 is binding to G quadruplex regions. However, the findings from the CLIP-seq data showing an interaction between NUP153 and viral RNA would be strengthened by performing an orthogonal assay, such as a gel-shift assay with viral RNA regions in question.

10. The hypothesis that NUP153 can modulate relative amounts of viral protein translation is intriguing. However, there is not sufficient evidence provided to make this claim and there is no data that support the proposed mechanism by which NUP153 might regulate translation. Further, changes in relative levels of E/NS1/NS2b are quite modest and representative IF images for TBEV NS1 and E are not provided. Additional experimentation, including a viral protein translation and/or a minimal system with reporters adjacent to the putative NUP153 binding site to demonstrate an effect on translation is required. Evidence that a truncated polyprotein is produced, dependent on the presence of NUP153, would also be more convincing.

Minor Comments

1. In Figure S2F, why does the addition of auxin at 6 hpi lead to an increase in the percentage of infected cells? This should be addressed.

2. Several different cell types are used throughout the study, which may partially explain differences in the magnitude of phenotypes. The addition of more relevant cell types would also be informative.

3. The authors do not provide a justification as to why the aa 1-650 construct was used for the CLIP-seq experiment.

4. Line 178: the authors claim the interaction between Nup153 and NS3/NS5 upon RNase treatment is evidence that the interaction between these proteins is direct. This experiment does not support that conclusion, only that the interaction

between Nup153 and NS3/NS5 is not dependent on RNA. The authors more accurately describe this result in lines 195-196 but should clarify in the previous statement.

Version 1:

Reviewer comments:

Reviewer #1

(Remarks to the Author)

In their updated manuscript, the authors have provided significant additional data to support their claims. This has addressed many of my concerns and increased overall confidence in the findings. However, there is still some uncertainty as to the specific role of Nup153 in flavivirus infection. Although the authors provide compelling evidence that Nup153 binds to viral RNA through its RBD and that Nup153 binding to viral RNA alters *in vitro* translation, the role of Nup153 within infected cells is less clear. Based on their data, the authors conclude that Nup153 is altering the ratio of viral structural and non-structural proteins, leading to increased assembly. Additional evidence is needed to support a role for Nup153 in virus assembly and to conclude that this effect is due to translational changes in the polyprotein. Specifically, when Nup153 is depleted starting at 6 hr after infection (should be completely degraded by 8 hr post-infection), no effect of infected cells or virus titers was observed. On the other hand, when BFA is added at 16 hr post infection there is a complete loss of virus production. If the effect of Nup153 depletion is specifically on virus assembly, there should still be a strong effect even when Auxin is added after 6hr, similar to BFA. Rather, these data suggest that Nup153 may act upstream of assembly. An effect on genome replication or general polyprotein translation efficiency could also lead to decreased spread observed in Figure 8. To make the claim that Nup153 is acting to specifically promote assembly, the authors should rule out an effect on virus replication. Without this direct evidence, it is challenging to determine if the minimal changes in protein ratios observed in Figure 8 are leading to a significant change in assembly. In the absence of additional experiments, their conclusions should be adjusted. Several specific points are indicated below.

-What is the level of virus spread at 16hr in Nup153(NG)aid cells? In fig. 2, the authors show that Nup153 depletion leads to a decrease in the number of infected cells, viral RNA and titers. It is not clear if this effect is due to a decrease in virus spread or a decrease in virus replication. The observation that the impact on virus infection is completely lost when you deplete Nup153 6 hr after infection suggests that virus replication and not infection is affected.

-To rule out an impact on virus replication, the authors should evaluate virus replication using subgenomic replicons or in a system where there is limited virus spread.

-Does Nup153 overexpression increase the structural to non-structural protein ratio in cells? Considering the proposed model, overexpression of Nup153 in infected cells should shift this ratio in the opposite direction from the depletion of Nup153.

Reviewer #2

(Remarks to the Author)

In this revised version, Peters and colleagues have included many experiments that answer several important concerns I raised on the first version. Notably, the new data based on super-resolution microscopy as well as the titration assays convincingly demonstrate that NUP153 accumulate in the replication compartment and is a proviral factor. In that sense, the manuscript has been much improved. A lot of nice work has been done.

However, the main mechanistic conclusion regarding the regulation of viral translation by NUP153 depending on the coding sequence remains not convincing in my opinion. It seems indeed that there is a mild change in the abundance of E compared to NS proteins as now shown by mass spectrometry and immunoblotting. Nevertheless, the *in vitro* translation assays are not enough to explain the phenotypes. Even if this set of data suggest that NUP153 RBD impact translation in a E sequence dependent manner *in vitro*, it remains very hard to comprehend how the rate of translation would be changed when the ribosomes translate the sequences encoding non-structural proteins as part of the same open reading frame. In line 64 and 475, the authors mention that the results "indicate NUP153 involvement in translation elongation" but this not addressed in this study per se. Ribosome profiling would be the gold standard if the authors want to make such a conclusion. So far, the data only indicated that site 1 in E impact translation *in vitro* when NUP153 RBD is added. It might also be more complex in cells, when the ER is present and remodelled by the virus. Without additional experiments addressing translation elongation, the conclusion is not supported by the data in my view. It is more likely that NUP153 impact the viral protein abundance at the posttranslational level, which was not addressed. The half-life the viral proteins could be tested by western blotting in wt and NUP153-depleted cells. This should not change according to the model elaborated by the authors.

Other comments:

-In Fig 4D, even if the data have been normalized to the input, these experiments tend to show that NUP153 associate with the viral negative strand (actually in a similar extent as the positive strand if the upper data point is not considered in the far right series). Even if the negative strand was not detected in the CLIP-seq data, the conclusion about the exclusive binding to the positive strand should be tuned down.

- The quality of Fig S7D is not optimal, and the results are hard to appreciate for the reader.

-Fig S8C is a bit confusing. What is the difference between the multiple "+" lanes? Are they biological replicates?

Reviewer #3

(Remarks to the Author)

The authors have addressed the majority of my concerns and added significant new experimentation and compelling evidence to support their claims. There are some outstanding questions/comments to address:

1. There is not a significant explanation as to why auxin addition after 4 hpi would have no impact. Why would this mechanism not be effective later in the infection cycle, closer to when C-prME are required for virion assembly?
2. The data suggests a mechanism controlling translation; however, there are no data examining differences in protein stability. This caveat could be addressed in the text.
3. In Figure 4G, RNA binding may be somewhat dependent on the protein expression levels of the various truncated constructs. Expression of eGFP seems to be variable by IF in Figure 4B. Is RNA binding normalized to protein expression? A western blot showing relative protein expression of the constructs should be included.
4. The paragraph explaining the binding peaks and the mutants generated is difficult to understand (lines 251-267). A more clear explanation would be helpful.
5. Figure 2G – while the interaction between NUP153 and Capsid is diminished with RNase treatment, it is not abolished as stated in the manuscript (Lines 179-181). Further, it appears that RNase treatment results in an overall decrease in capsid expression in the lysate.

Version 2:

Reviewer comments:

Reviewer #1

(Remarks to the Author)

The additional data provided by the authors significantly enhance confidence in their conclusions, and the revisions to the manuscript satisfactorily address my major concerns.

Reviewer #2

(Remarks to the Author)

In this revised version of the manuscript, Peters and colleague have notably complemented the study with data showing stability of viral proteins over 6 hours and replication kinetics in wt and NUP153-deficient cells. Unfortunately, while these data are informative, they do not contribute to firmly conclude that NUP153 regulates the translation of structural proteins to favor viral particle production, the main claim of the work. Although all aspects related to elongation have been removed from the text, a potential role of NUP153 on translation still only relies on in vitro assays and reporter constructs far from the reality of a flaviviral genome. Thus, my previous concern remains, especially since the new results from stability assays as such (Fig 8 and Suppl Fig 8) do not allow to measure the half-life of the viral proteins as they look quite stable over the 6 hour time frame. Again, direct evidence should support this claim such as for instance: 1- evaluating vRNA distribution change in the fraction of a polysome sucrose gradient, or 2- viral protein/reporter expression following the transfection of a replication-deficient full-length viral genome (which does not require high transfection efficiency).

Reviewer #3

(Remarks to the Author)

The authors have provided sufficient experimentation to address the majority of reviewer concerns. Additional studies, including replicon-based translation assays and more in-depth work to quantify viral protein half-life in the context of infection would further support a role for NUP153 in specifically regulating translation. However, this is an informative study that convincingly shows that NUP153 is a pro-viral factor in flavivirus infection, which impacts the ratio of structural to nonstructural proteins and posits an intriguing model for translational control in flavivirus infection.

Point by point response to reviewer, answers are marked in red.

Reviewers' comments:

Reviewer #1 (Remarks to the Author):

Peters et al present evidence that the nuclear pore complex protein Nup153 is a proviral factor that promotes flavivirus infection. With a specific focus on tick-bore flaviviruses (TBEV and LGTV) the authors show that Nup153 is upregulated and re-localized to the cytosol during infection and that depletion of Nup153 leads to a decrease in the number of infected cells. Furthermore, they present evidence that Nup153 interacts with viral NS3 and NS5 proteins and suggest that these interactions facilitate Nup153 recruitment to the cytosol. The authors also show evidence that Nup153 interacts with viral RNA through an RNA binding domain at several G-rich regions of the viral Genome. Finally, they use light microscopy to evaluate the levels of viral structural and non-structural proteins in Nup153 depleted cells and report a change in the ratio between E and NS2B. Based on these data, the authors posit that Nup153 is involved in promoting the translation of structural proteins to facilitate virus assembly and spread. In general, the study is well structured, using various techniques to comprehensively evaluate the role of Nup153 in flavivirus infection.

We thank the reviewer for the positive feedback and for appreciating the study..

My major concern with the current study is that the overall phenotype of Nup153 re-localization and the effects of Nup153 depletion on virus infection are modest. Considering that the authors also note a significant decrease in cell proliferation upon Nup153 depletion (Figure 3), it is possible or even likely that Nup153 may not have a direct role in virus infection. Rather, Nup153 may be required to maintain other cellular pathways that are needed for infection. Finally, the current data do not sufficiently support a role for Nup153 in viral translation. Substantial further experimentation is needed to support the author's conclusions.

We have addressed the raised questions in a point-by-point manner and added experimental results that further support the conclusion.

Specific points:

1.Previous reports have shown that NUP153 is elevated in nuclear and cytosolic fractions

during stress responses (PMID: 20041180). Therefore, the re-localization of Nup153 may not be virus-specific but is caused by the upregulation of Nup153 which leads to increased cytosolic protein levels that localize to the ER. Therefore, the authors need to provide sufficient controls and parallel validations to confirm the specificity of their results for virus infection. For example, ER general markers for microscopy experiments and ER resident specificity control proteins for IP experiments.

We thank the reviewer for the suggestions.

We have restructured figure 1 and added super resolution microscopy Structural illumination microscopy (SIM) to show that NUP153 localizes together with the ER marker Calnexin during infection (see new figure 1D) together with quantification of the Manders coefficient (new Fig 1F), which answers the question. Moreover, we have also analyzed the upregulation of NUP153 in MAVS^{-/-} ko A549 cells which are unable to upregulate IFN β and showed that NUP153 is upregulated after infection independent of type I IFN response (new figure 1H-J).

2. There is some discrepancy between the systems for Nup153 depletion in Figure 2. For the KO cells, there is a 20% reduction in Nup153 leading to a 70% reduction in infected cells while a complete depletion of Nup153 using the AID system only reduces infection by 50%. Also, while depletion experiments are performed for both LGTV and TBEV using the AID system, the KO experiments are only done for LGTV and not TBEV.

We thank the reviewer for pointing out this discrepancy. To avoid confusion, we have removed the KO cells and only use the results of the much cleaner AID system, which makes the manuscript clearer. We have further focused figure 2 on TBEV.

3. Throughout the manuscript, the timing and MOIs for infection experiments as well as the use of LGTV vs TBEV in specific experiments are not well justified. Considering that the authors are reporting an effect specifically at early timepoints after infection they should include detailed information on the kinetics of virus infection (RNA levels and virus titers) over the course of infection. They should also be consistent, when possible, with infection times and MOIs. Some specific examples are listed below.

a. The authors use 16hr, 24 hr or 48 hr for TBEV, 24 h or 48 h for LGTV and 48 hr for all other flaviviruses without justification. These differences make it difficult to compare results between experiments.

Thank you for pointing this out. We have now reduced the number of cell types used in the study to make the message clearer. The MOI stated was calculated based on virus titrated on VeroB4, however the kinetics of virus infection is different in different cell types, so we have recalculated the MOI corresponding to the cell type in the specific experiments. The different time points correspond to 50 % infection rate and since the different viruses have different kinetics we needed to use different infection times. We have clarified these points in the material and method and also in the figure legends.

b. There are significant differences in MOI used for experiments with no explanation as to why. For example, the virus spread experiment in Figure 6G was done with an MOI of 0.05 for TBEV and an MOI of 5 for WNV. For this example, it is not clear how a virus spread experiment can be performed with a starting MOI of 5 which should infect nearly all cells with the initial infection. Similar MOI discrepancies are seen throughout the manuscript and should be addressed.

Thank you for pointing this out originally the titers of the stock was calculated on the VeroB4 cells, however as mention in the previous point we have recalculated the actual infectious dose by titrating the virus on the specific cell lines.

c. In Fig 2 the analysis of LGTV and TBEV is inconsistent. Time of addition experiments for LGTV should also be shown as well as FFU assays for all time points. These data would be useful for identifying the specific effects of Nup153 depletion timing.

We agree. We have added FFU for both TBEV and LGTV in figure 2. However, as we now focus the manuscript on TBEV we are not adding time of addition of LGTV.

4. The authors state that Nup153 is localized to the “amplification site” and not “replication vesicles” based on images and quantification in Figure 1H-I. First, the differences between these two regions are not well defined. Second, since previous reports have shown that replication vesicles form within rough-ER sheets that also support translation (see PMID: 26073783, 28249158, 19380115) it is not clear how sites of replication and amplification can be resolved using confocal microscopy.

We have performed super resolution microscopy SIM to clarify if NUP153 co-localize with dsRNA which is a marker for the replication vesicles. Most of the dsRNA colocalize with NUP153, although not all NUP153 localize to the dsRNA (New figure 1D-G). This indicate that NUP153 does not target the replication vesicle specifically but rather is in the ER, which

we also show through the colocalization with the ER marker Calnexin. We choose to use “amplification site” as a more general term instead of “replication vesicles”, as the resolution of the SIM is not high enough to show that NUP153 is inside the vesicles.

5. The function of Nup153 in promoting structural protein translation is not well supported by the data. The authors see only a very modest change in the levels of 2 viral proteins (E and NS2B) which could be explained through changes in cell growth or protein stability. These findings should be validated by western blot and additional structural and non-structural proteins evaluated.

Thank you for this comment. We have worked extensively on this point and generated several figure panels (see Figure 8). Regarding the immunofluorescence data on the ratios of E against NS1/NS2B, we opted to use microscopy so that we could calculate protein levels exclusively in infected cells (Figure 8E-G), and we clarified this in the materials and methods section. Additionally, we have added WB of AID cells infected with TBEV (Figure 8H and Supp Figure 8C). Using the available Ab for TBEV we could show that ratio C/E and M/E does not change but NS1/E, NS2B/E, NS3/E and NS5/E increases (Figure 8H). Also as seen by the tubulin staining in these Western blots when cells are seeded to be confluent as done for all infection assays NUP153 depletion has minimal to no effect on number of cells (Supp Figure 8C). Since we do not have the required set of antibodies against WNV we choose to do mass spectrometry on the WNV infected cells and found that C/E, prM/E does not change but the ration NS1/E, NS3/E and NS5/E do increase (New supp Fig 8E and Fig 8I). These data support our original claim. We have also added in vitro translation in the presence of the RNA binding domain and found that by adding NUP153-RBD the downstream protein production is reduced new figure 8D.

6. Further to the previous point, the authors identify an interaction between Nup153 and viral RNA, but it is not clear if Nup153 binding is specific to viral RNA or just general RNA binding activity. Less than 1.4% of the reads in the CLIP-Seq were associated with viral RNA at a time point where the viral RNA should be highly abundant. If Nup153 is significantly impacting translation, the amount bound to viral RNA at this point in infection should be much higher.

NUP153 is an RNA binding protein that is involved in the transport of RNA (and proteins) through the nuclear pore, and still has its normal function during infection, in addition to binding to viral RNA.

The virus is not affecting the RNA transcription of cellular RNA.

Consistent with other studies (PMID: 31384002; PMID: 40161615) we find that only a few % of the binding RNA is viral RNA during infection. We further show that NUP153 as expected bind to cellular RNA, as well as very specifically to the motif identified in the virus genome. The suggested binding site in the viral genome is very specific. To support this, we have now added RNA EMSA experiments of NUP153-RBD and different RNA constructs that clearly show that the RBD binds with higher affinity to the peak identified with clip-seq as compared to mutated version of the peak (Figure 5E-H).

7. In Figure 4, the relevance of the Nup153 RBD for infection should be validated with RBD specific deletions in the full-length Nup153. Rescue experiments should also be done to show that expression of RBD containing Nup153 variants can rescue Nup153 depletion during TBEV or LGTV infection.

Thank you for this helpful suggestion. We have added the a truncated constructed of Nup153 lacking RBD and found that it does not bind to the viral RNA in the Clip assay, and it can also not rescue the infection (see new Figure 4C and F).

Minor comments

- The CLIP sequencing experiment in Figure 4 shows several other peaks including a strong peak in the 3'UTR that is not discussed.

This peak is not significantly enriched in all three replicas when using the PEAKachu algorithm and therefore they are not evaluated further.

- The authors state that the interaction between Nup153 and C is RNA dependent (Fig3H) but there appears to be a loss of C in the input when RNase is added in all cases.

This is true, C protein of flaviviruses is known to be extremely positively charged which means it can bind to negatively charged RNA through electrostatic interactions. Using RNase to degrade the RNA could therefore alter the C proteins stability or ability to efficiently migrate in a gel.

- Several of the western blots show multiple species for Nup153 without explanation or molecular mass indicators.

Yes we do acknowledge this, and the band with higher molecular weight corresponds to full length NUP153-NG-AID construct, however NUP153 does have other isoforms that could

potentially be detected with our antibody and be more pronounced in the NUP153 [NG]^{AID} cells compared to their parental DLD cell line.

Reviewer #2 (Remarks to the Author):

In this study, Peters and colleagues have studied the relationship between the nuclear pore protein NUP153 and flaviviruses (mostly tick-borne viruses (TBEV and LGTV)). They show that this host factor is overexpressed in infected cells and is relocalized to the cytoplasm although the latter must be analyzed further. The authors nicely show that NUP153 binds to the viral RNA and that its protein interactome is modulated upon infection. Since NUP153 is required for cell survival, the authors took advantage of an inducible system to rapidly degrade NUP153 and assess its role during viral infection. They further propose that NUP153 regulates flavivirus life cycle at the translational level by changing the expression ratio between structural and non-structural proteins. Unfortunately, as explained below such model is not supported by the data in my opinion since no translation assays was performed. Furthermore, the requirement of NUP153 for TBEV replication remains to be confirmed.

Major comments:

1- The included data of Figure 1 were not entirely convincing regarding a relocalization of NUP153. Indeed, as shown in Fig 1D-E, NUP153 is overexpressed in LGTV and TBEV-infected A549 cells at both protein and mRNA level. Thus, the observed cytoplasmic signal in Fig 1A could simply be due to the fact that the sensitivity is higher. Actually, the nuclear signal increases as well in the same order of magnitude. To claim that there is a relocalization of NUP153 to the flavivirus replication compartment, the cytosol signal/nuclear signal ratio is expected to increase and should be determined in Fig 1A-C and 6A-B. According to the data of Fig 1C, it seems not to change much. Thus, the apparent cytoplasmic signal would be due to a global overexpression, and this would not be a relocalization process. In that case, it would only demonstrate that NUP153 colocalizes with NS3. To support their conclusion, the authors have performed sub-cellular fractionation, and the resulting quantification showed a significant increase of NUP153 in the endoplasmic reticulum (ER) fraction. However, the corresponding immunoblotting are importantly missing and must be shown. In addition, markers of the nucleus, mitochondria, ER and vesicles should be used to ensure that the fractionation process was actually efficient in separating the organelles.

We thank the reviewer for the comment and have decided to remove the fractionation. We instead added super resolution microscopy to show the relation between the viral proteins,

NUP153 and ER marker Calnexin. Here we show that the staining of NUP153 and Calnexin overlaps almost completely. We have calculated the ratio between the signal in the nucleus and cytoplasm and there is no difference between uninfected and infected cells this data has been added as Supp Fig 1A.

Fig 1H clearly shows some colocalization between NUP153 and dsRNA. This is not consistent with the quantification shown in Fig 1I. Such colocalization was expected in my view given that NS3 mostly associates with replication complexes and that NUP153 associates with both the minus (-) and plus (+) strands of viral RNA (Fig 4B). This is not consistent. In the same line of idea, the authors conclude with Fig 4B that NUP153 preferentially associates with the (+) strand, but the data show that it also associates with the (-) strand. Such binding is not trivial given that there are many more copies of (+) strands than (-) strands in the replication compartment (which is not measured here). Considering this, Fig 4B would support the conclusion that NUP13 associates with replication complexes. In any case, the input (-) and (+) RNA levels (i.e., before IP) should be measured.

We thank the reviewer for this comment, and agree that this is important to clarify if NUP153 binds to positive or negative RNA strands, or both. We have chosen to remove the confocal microscopy image quantification in figure 1I and replaced them with super resolution microscopy and quantification of that new figure 1D-G. Here we show that most of the dsRNA colocalize with NUP153 but only a fraction of the NUP153 colocalize with dsRNA. NUP153 is rather localized to the ER as shown with the overlap in the calnexin staining.

For the CLIP assay the values are already normalized to the input for the negative and positive, we have now clarified this in the figure legend. No negative stranded RNA was detected in the CLIP Seq, suggesting that NUP153 does not bind to negative RNA. We potentially observe some binding to the negative strand RNA in the CLIP assay that could be a result of it indirectly being pulldown in the form of partial dsRNA, however based on our CLIP-seq assay we believe the interaction occurs on the positive strand RNA.

2- The potential proviral role of NUP153 needs to be confirmed. First, all the replication assays were performed by measuring the % of infected cells. This is not the best readout as it measures the capacity of the virus to spread in culture but not directly its replication efficiency. All the results in Fig 2H, 6C and 6G should be confirmed by plaque/FFU assays and/or RT-qPCR (ideally both) which provide a better dynamic window than an imaging/FACS approach. More importantly, Fig 2I is problematic since no reduction in particle production is observed, implying that NUP153 depleted cells are still able to resume a full life

cycle. This was measured at 6 hours post-auxin addition, a time point at which the phenotype of the % of infected cells is not obvious (Fig 2H). This should be performed when the treatment is optimal (e.g. addition at 2 hours post-infection). Furthermore, Fig 3F shows that NUP153 depletion reduces cell proliferation, which may be detrimental for viral replication in general, that is, not only because of the lack of NUP153 in the replication compartment. Without a clear impact of NUP153 depletion on vRNA levels and/or viral particle production (which both rely on efficient viral translation) for same number of infected cells, it cannot be concluded at that stage that NUP153 modulate flavivirus life cycle.

Thank you for suggesting. We have now the FFU for all these panels and time points, we have also transfected AID cells with NUP153 to rescue the infection, to confirm that NUP153 is important for infection Fig 2N and Fig 4F. Q-PCR measurements has been added Fig 2M.

We have removed the cell proliferation assay. In contrast to the proliferation assay, infection assays were seeded to obtain confluent wells, meaning that at end point similar number of cells can be found in the wells. Additionally, we show that NUP153 depletion does not affect viability, however as NUP153 is required for assembly of the nuclear pore complex it is essential for the replication of the cells.

3- Mostly based on NUP153-dependent differences in NS3, NS2B and E protein expression levels, the authors conclude that viral RNA translation is regulated by NUP153. However, this was never addressed in this study in proper translation-based assays. Moreover, as the authors mention in the discussion, the 10 structural and nonstructural proteins are produced from the same open reading frame as a polyprotein and hence, at an equimolar ratio following processing. So, it is really hard to envision how a modulation of translation would explain the changes in NS1/E and NS2B/E protein ratios upon NUP153 depletion. Since E coding region is upstream that of NS proteins (more abundant in depleted cells), how would that work in the absence of a stop codon? Would there be an alternative translation initiation site upstream the NS1 coding region (which requires E amino-terminus to be inserted in the ER membrane). It is much more likely that viral protein expression level is modulated at the posttranslational level. Moreover, in that specific case, quantifying viral protein expression using microscopy is not appropriate since some protein population might be less accessible to the primary antibodies without allowing the monitoring of protein molecular weight/maturation. This must be done using immunoblotting. Without addressing translation per se, the model proposed in Fig 6H is not supported and really hard to conceptually comprehend, especially since it goes against the current knowledge of flaviviral polyprotein production. Using a reverse genetics system of TBEV could help characterize mutated

genomes (which do not bind to NUP153) during translation and viral RNA replication in transfected cells.

We have invested significant efforts towards understanding the experimental observations. We have added WB of TBEV infected AID cells and calculated the ratio between C/E, M/E, NS1/E, NS2B/E, NS3/E and NS5/E (limited by the availability of antibodies). We did not see a difference in the ratio between C/E and M/E in the absence of NUP153 but the ratio of NS1, NS2B, NS3 and NS5 to E increased. In parallel we used WNV infection, and since the availability of Abs are limited, we used mass spectrometry instead (peptides were detected for C, prM, E, NS1, NS3, NS4B, NS5). Data were very similar between WNV and TBEV experiments supporting our original claim. We have also expressed the RBD from NUP153 and tested its effect on in vitro translation. Here we show that adding NUP153-RBD reduces luciferase (Luc) expression downstream of the RNA binding site, and increases the Luc signal if the RNA motif is downstream the Luc ORF. Additionally, WB analysis of the in vitro translation assay in the presence of RBD results in production of lower molecular weight protein that corresponds to C-prM-E.

Finally, we note that the RNA motif identified to bind to NUP153 is highly conserved among all flaviviruses. We constructed mutant viruses lacking the conserved Gs, however they were not viable even after adding E in trans.

Other comments:

- How is NUP153 overexpressed? Since these experiments were performed in interferon competent A549 cells, is this the consequence of the TBEV/LGTV-induced innate immune response? The authors mention MAVS knockout A549 cells in the Methods section. Is the infection-related induction of NUP153 expression also observed in these immune-deficient cells?

Thank you for this question. We have added a new panel fig 1H-J where we analyze if NUP153 upregulation after infection is dependent on IFN upregulation and we can show that it is not. We also show that overexpression of NS3 and NS5 expression plasmids results in increased expression of NUP153 (Supp fig 3B-C).

- Are the observed phenotypes specific to NUP153? Other NUPs should be tested for colocalization with viral proteins and potential pro/antiviral roles.

We agree. However, there are many NUPs and it is out of the scope of this paper to test all of the NUPs for their function during Flavivirus infection. We are working on some other NUPs and their function during TBEV infection.

- Line 41: “, and virion assembly we define...”. Please rephrase **Ok**
- Line 51: “has is required”. This should be rephrased. **Ok**
- Line 52: To my knowledge, HCV does not induce convoluted membranes. Moreover, reference 15 relates to a study on HIV and not HCV. **We have changed it to membranous web and removed the reference**
- Line 53: “similar but different...”. This reads strange. **Rephrased**
- Line 88: NS3 is not located in the the ER (i.e. in its lumen) but is anchored to its cytosolic side by interacting with its co-factor NS2B. **Has been clarified.**
- Line 160: Examples of identified proteins involved in cell cycle regulation should be indicated. **Ok**
- Fig 3H: Why is there a shift in the size of C in the input upon RNase treatment?

C protein of flaviviruses is known to be extremely positively charged and bind to RNA in an electrostatic dependent manner. Treating samples with RNase could alter the protein stability or ability to efficiently migrate through a gel.

- Line 178 : The fact that RNase treatment did not disrupt the interaction between NUP153 and NS3 or NS5 does preclude that it is indirect (e.g., mediated by other proteins). This sentence must be rephrased. **done**
- Line 180: “flaviviruses” (typo) **ok**
- Fig 4E: The input RNA level should be measured to determine whether the -strand/+strand ratio is the same in the IP. **It is normalized to the input has been added to the figure legend.**
- Fig 5: Is there a change in the binding of NUP153 to cellular mRNAs. Such analysis of the data acquired in the CLIP would significantly strengthen the study and would be consistent with the observed change in NUP153 protein interactome.

This is a good suggestion, however, the Clip-seq data does not have enough depth to do this analysis. We looked at the binding of NUP153 to Actin RNA with Clip assay and found that during infection NUP153 bind actin RNA Supp fig 4A.

- Line 370: This conclusion is not supported by the data as there is no experiment demonstrating the importance of the conserved G-rich region during replication. This statement must be tuned down. **Rephrased**

- Line 402: This statement must be rephrased as the observed phenotypes on viral protein abundance could be very well due to a posttranslational regulation.

We respectfully disagree. We have provided additional data to support it, see the previous comments.

Reviewer #3 (Remarks to the Author):

The manuscript submitted by Peters, et al. is an interesting study suggesting that a nuclear pore protein, NUP153, is a proviral factor for TBEV and several other members of the Orthoflavivirus genus. The authors suggest that NUP153 is co-opted during flavivirus infection and recruited from the nucleus to the ER, where it regulates viral polyprotein translation. The data presented convincingly demonstrates that NUP153 is upregulated in response to tick-borne flavivirus infection and suggests some relocalization from the nucleus to the ER upon infection. Further, the authors present evidence showing that the RNA binding domain of NUP153 is required for relocalization and that NUP153 interacts with flavivirus RNA at G-quadruplex structures. The authors go on to suggest that NUP153 regulate translation of the flavivirus polyprotein via the interaction with viral RNA at the C-terminal portion of the E protein coding sequence, resulting in a relative increase in the translation of the upstream structural proteins, as compared to the downstream nonstructural proteins. While this is a novel and compelling model, the data presented in the manuscript do not sufficiently support these claims. Further, the addition of controls and orthogonal approaches to bolster these data are required to support many of the other conclusions.

Thank you for your analysis, we have now added several new orthogonal assays to support our claims.

Major Comments

1. In Figure 1A, the immunofluorescence images presented to suggest co-localization and re-localization require controls for proper interpretation. The overall signal of NUP153 appears to increase everywhere in the cell, which supports upregulation of this protein during infection. However, the evidence that abundance is increasing at viral replication sites is not convincing and requires further experimentation, including cellular fractionation with appropriate cellular component controls. Both an ER marker and another nuclear envelope

protein should be included in these experiments and in IF experiments throughout the manuscript.

Thank you for the suggestion. We show that NUP153 increases both in the nucleus and the cytoplasm after infection but that the ratio stays the same (see new Supp Fig 1A). As it is very difficult to fractionate cells without contamination between the different organelles, we removed the fractionation data and analyzed the localization of NUP153 with super resolution microscopy. In these experiments we used calnexin as an ER marker, NS3, and dsRNA for virus and calculated the colocalization between the different staining (see New figure 1D-G). The integrity of the nuclear envelope after infection was analyzed by FG binding Ab Mab 414 (see Supp fig 1 C-D).

2. Similarly, markers for all compartments should be included in fractionation experiments (Figure S1E). It is possible that the integrity of the nuclear envelope compromised during infection, resulting in a change in localization of nuclear proteins. Why are calnexin and GAPDH present in the pellet of all fractions in uninfected cells? This suggests the fractionation of specific compartments is not robust. Additional cellular markers, including mitochondrial and another nuclear marker should be included in this experiment. Further, the quantification of the IF data in Figure 1C suggests an increase in NUP153 in the nucleus but the quantitation of the fractionation data in Figure 1G does not reflect this. This discrepancy should be addressed.

We agree with the reviewer. It is very difficult to fractionate cells and separate different cellular compartments, as nucleus and ER are connected, and mitochondria and ER are connected, we decided to remove this set of data. We instead chose to add super resolution microscopy to have a clearer picture of the NUP153 protein localization.

3. Figure 1D does demonstrate an increase in NUP153 RNA and but the increase in protein is rather modest and does not appear equivalent to the increase in signal shown in Figure 1A. Further, the authors claim that this data distinguishes between increased expression and increased stability of NUP153. As NUP153 protein stability assays have not been performed here, this claim is not supported.

Thank you for pointing this out. We have now rephrased our findings to only state that we observe an increase in NUP153 expression on RNA and protein levels. The NUP153 RNA levels after infection in 293 cells are greater, and this information has been added to the supplementary (Supp Figure 1B). We have also added a reference about the stability of NUPs (PMID: 23993091 and PMID: 35412236).

4. The transient knock-down of NUP153 is very inefficient, despite being performed twice and sorting on GFP positive (Cas9 positive) cells. While there is a remarkable decrease in infection in these cells, it is difficult to compare to the control (GBF1) as no knock-down data is provided here. Transfection of siRNA or shRNA targeting NUP153 would also provide transient depletion at increased efficiency. Further, this would also allow for additional infection assays, such as quantitation of FFU or PFU in the cells to further support a proviral role for NUP153 and viral life cycle assays to supplement the findings from the time of addition assays.

As the NUP153 protein half-life is long (PMID: 35412236) it affects all systems of transient knockdown that only reduces the newly synthesized NUP153. We have removed the knockdown data and focused on the more clean AID degradation system. We have also shown that NUP153 overexpression can rescue the effect of NUP153 degradation.

5. The NUP153(NG)AID system appears very robust but the infection phenotype is less pronounced here than in the transient knock-down experiment (Figure 2B) despite the complete depletion in AID system. This suggests that the phenotype in Figure 2B is not solely dependent on NUP153. The expression of NUP153 post-auxin addition should be shown for all time points. After the addition of auxin, at what point does expression of NUP153 recover? Additional viral infection assays, including the FFU assay should be included for all time points, not just the 6h timepoint at which there is no effect on infection.

Thank you for your comments, we have added FFU for all time points. In the experiments using the AID cells the Auxin is present during the whole infection time so there is no recovery of NUP153. We have also seen that the effect of NUP153 degradation on progeny particle production is greater compared to number of infected cells.

6. In general, the description of experimental design, assays, normalizations and reagents used is insufficient. For several experiments, it is not clear what is being used as a negative control nor what the assays are precisely measuring. For example, in Figure 2A, data is normalized to CAS9. Is this Cas9 alone or with the non-targeting guide RNAs? Similarly, a description of the NUP153(NG)AID and control cell lines used needs a clearer description to be able to adequately interpret the results.

This is a valid point. We have removed the CAS9 data in favor of the more robust AID system. We have also added a description of the AID cells in the results section.

7. In Figure 3, the differences in NUP153 interactors in mock vs. infected cells is interesting and potentially informative for function. However, there are no experiments validating these differential interaction results.

We agree that this is interesting however, this is out of the scope of this manuscript.

8. In Figures 4C-E, another unrelated RNA binding protein should be used as a control. Additionally, a non-viral RNA should be quantified in 4C-4E to show specificity of interaction with viral RNA.

We understand the reviewer's concern. However, as to our knowledge, it is not known what RNA binding proteins of the cell are not binding flaviviral RNA, therefore we are unable to do this control. If the reviewer has a suggestion of an unrelated cellular RNA binding protein known not to bind flavivirus RNA we would be happy to use it as control. Furthermore, most RNA binding proteins have RNA binding domains that are positively charged, and RNA being negatively charged in general will always result in some level of binding. However, the added RNA-EMSA assay show that the RNA binding domain of NUP153 binds better to the identified peak than the mutated constructs. We have also added RNA-EMSA using purified GFP protein to show that the shift that we observe is specific to the RNA binding domain of NUP153. In addition, it is difficult to choose a cellular RNA that NUP153 does not bind to, as one of the functions of NUP153 is to bind to cellular mRNA. We also show now that NUP153 bind to Actin mRNA in a Clip assay new Supp figure 4A.

9. The results in Figure 5 are well-done and suggest that NUP153 is binding to G quadruplex regions. However, the findings from the CLIP-seq data showing an interaction between NUP153 and viral RNA would be strengthened by performing an orthogonal assay, such as a gel-shift assay with viral RNA regions in question.

Thank you for this suggestion. We have now done RNA EMSA experiments on full length viral RNA, different fragments and mutants. Our new data strongly show that NUP153 RBD specifically bind the conserved G in Peak 1.

10. The hypothesis that NUP153 can modulate relative amounts of viral protein translation is intriguing. However, there is not sufficient evidence provided to make this claim and there is no data that support the proposed mechanism by which NUP153 might regulate translation. Further, changes in relative levels of E/NS1/NS2b are quite modest and representative IF images for TBEV NS1 and E are not provided. Additional experimentation, including a viral protein translation and/or a minimal system with reporters adjacent to the putative NUP153 binding site to demonstrate an effect on translation is required. Evidence that a truncated

polyprotein is produced, dependent on the presence of NUP153, would also be more convincing.

Thank you for these suggestions. We have now added in vitro translation experiments adding GFP or GFP-NUP153-RBD and find that translation (luciferase activity) after Peak 1 decreases after addition of the RBD and that luciferase activity increases when peak 1 is after the Luc ORF, new fig 8 A-D. Additionally, we show that the addition of the RBD does not result in no translation, resulting instead in production of a lower molecular weight product positive for E, further supporting our claims.

We have also run WB of several structural proteins and non-structural protein of TBEV and shown that the ratio of C/E, prM/E stays the same whereas NS1/E, NS2B/E, NS3/E and NS5/E ratios increases in the absence of NUP153. For WNV as we do not have antibodies we instead used mass spectrometry and found similar results new figure 8H-I, supp fig 8C-E.

Minor Comments

1. In Figure S2F, why does the addition of auxin at 6 hpi lead to an increase in the percentage of infected cells? This should be addressed.

We do not know why there is an increase after 6 h, however, all the figures after this Auxin was added 2 hours prior to infection, so it is not influencing the main findings. We also feel that this observation is out of scope of this manuscript which focuses on NUP153.

2. Several different cell types are used throughout the study, which may partially explain differences in the magnitude of phenotypes. The addition of more relevant cell types would also be informative.

We see the same phenotype in A549 cells, A549 MAVS^{-/-}, 293T cells and DLD1 cells, with all flaviviruses tested.

3. The authors do not provide a justification as to why the aa 1-650 construct was used for the CLIP-seq experiment.

Thank you for the comment. As the FG region of NUP153 is known to bind many different cellular proteins, we used the truncated version 1-650 lacking this unstructured protein binding hub region to increase the specificity of the CLIP assay. Line 260 "We opted to use the eGFP-NUP153₁₋₆₅₀ instead of the full-length construct to minimize indirect RNA binding

arising from proteins bound to the FG-domain of NUP153 (Supp Fig 4C).” Additionally, we show in Fig 4F that this construct is able to be found at the amplification site, bind RNA and rescue infection.

4. Line 178: the authors claim the interaction between Nup153 and NS3/NS5 upon RNase treatment is evidence that the interaction between these proteins is direct. This experiment does not support that conclusion, only that the interaction between Nup153 and NS3/NS5 is not dependent on RNA. The authors more accurately describe this result in lines 195-196 but should clarify in the previous statement.

We have clarified it in the text.

REVIEWER COMMENTS

Reviewer #1 (Remarks to the Author):

In their updated manuscript, the authors have provided significant additional data to support their claims. This has addressed many of my concerns and increased overall confidence in the findings. However, there is still some uncertainty as to the specific role of Nup153 in flavivirus infection. Although the authors provide compelling evidence that Nup153 binds to viral RNA through its RBD and that Nup153 binding to viral RNA alters in vitro translation, the role of Nup153 within infected cells is less clear. Based on their data, the authors conclude that Nup153 is altering the ratio of viral structural and non-structural proteins, leading to increased assembly. Additional evidence is needed to support a role for Nup153 in virus assembly and to conclude that this effect is due to translational changes in the polyprotein. Specifically, when Nup153 is depleted starting at 6 hr after infection (should be completely degraded by 8 hr post-infection), no effect of infected cells or virus titers was observed. On the other hand, when BFA is added at 16 hr post infection there is a complete loss of virus production. If the effect of Nup153 depletion is specifically on virus assembly, there should still be a strong effect even when Auxin is added after 6hr, similar to BFA. Rather, these data suggest that Nup153 may act upstream of assembly. An effect on genome replication or general polyprotein translation efficiency could also lead to decreased spread observed in Figure 8. To make the claim that Nup153 is acting to specifically promote assembly, the authors should rule out an effect on virus replication. Without this direct evidence, it is challenging to determine if the minimal changes in protein ratios observed in Figure 8 are leading to a significant change in assembly. In the absence of additional experiments, their conclusions should be adjusted. Several specific points are indicated below.

Thank you for the insightful suggestions and comments. We agree that it is hard to pinpoint the specific effect of NUP153 on viral infection. We have therefore generated more data to clarify this. With the new data we see that early viral replication is not affected by depletion of NUP153 up to 12 h post infection. These results support our claim that NUP153 is not involved in replication per se.

New data Fig 2L and lines 130-135, “Because we detected differences between the reduction in number of infected cells compared to the released viral particles following depletion of NUP153 we measured the amount of TBEV RNA at 2, 8, 12 and 16 h post infection in cells induced with auxin 1 hpi (Fig 2L). We found that depletion of NUP153 led to approximately 50% reduction in TBEV viral RNA at 16 hpi, however, no reduction at 8 or 12

hpi (Fig 2L), indicating that NUP153 depletion does not affect early replication, but more likely affects virus assembly and release.”

We also acknowledge that the direct effect of BFA on particle release has a different phenotype compared to NUP153 depletion. This might be due to BFAs effect on viral replication if administered early in infection (DOI: 10.3390/v9010005). We choose to administer BFA after the viral replication complexes has formed to isolate the effect to particle release, however, the BFA remains in the supernatant until 24 hpi thus blocking infection of virions released into the supernatant before administration.

-What is the level of virus spread at 16hr in Nup153(NG)aid cells? In fig. 2, the authors show that Nup153 depletion leads to a decrease in the number of infected cells, viral RNA and titers. It is not clear if this effect is due to a decrease in virus spread or a decrease in virus replication. The observation that the impact on virus infection is completely lost when you deplete Nup153 6 hr after infection suggests that virus replication and not infection is affected.

Thank you for this suggestion. The level of infection in the NUP153 (NG) at 16hpi in Fig 2 is approximately 50%. To clarify in more detail in what stage the NUP153 is important for viral infection, we have performed additional infection experiments. NUP153 depleted cells and control cells were infected and TBEV RNA levels in the cells were quantified at different timepoint post infection (2, 8, 12 and 16 h post infection) and normalized to the 2 hpi time point. The 8 and 12 hpi time-point corresponding to when virus is replicating but not secreting new virions, and the 16 h post infection (corresponding to virus release). These data now replace the old Fig 2M. The results indicate that NUP153 depletion does not affect viral replication as measured at 8 and 12 h post infection, Lines 130-134.

-To rule out an impact on virus replication, the authors should evaluate virus replication using subgenomic replicons or in a system where there is limited virus spread.

Thank you for the suggestion, as these DLD NUP153 AID cells are very difficult to transfect with subgenomic replicon we opted to analyze viral replication at early timepoints when virion release has not yet been initiated. These data are now in the Fig 2L, we did not see any effects of NUP153 depletion up to 12 hours post infection.

-Does Nup153 overexpression increase the structural to non-structural protein ratio in cells? Considering the proposed model, overexpression of Nup153 in infected cells should shift this ratio in the opposite direction from the depletion of Nup153.

Thank you for this suggestion we have performed overexpression and analysis of protein ratio in transfected 293 cells and we do not see any changes when overexpressing NUP153. The reason for this is not clear, but one hypothesis could be that the expression level of NUP153 is already increased during infection Fig 1J, thus the additional overexpression of NUP153 with plasmids might not affect viral replication. In support of this, we show that the viral infection does not increase upon transfection of NUP153-eGFP in the no Auxin treated DLD NUP153 AID cells (Fig 2M).

Reviewer #2 (Remarks to the Author):

In this revised version, Peters and colleagues have included many experiments that answer several important concerns I raised on the first version. Notably, the new data based on super-resolution microscopy as well as the titration assays convincingly demonstrate that NUP153 accumulate in the replication compartment and is a proviral factor. In that sense, the manuscript has been much improved. A lot of nice work has been done. However, the main mechanistic conclusion regarding the regulation of viral translation by NUP153 depending on the coding sequence remains not convincing in my opinion. It seems indeed that there is a mild change in the abundance of E compared to NS proteins as now shown by mass spectrometry and immunoblotting. Nevertheless, the in vitro translation assays are not enough to explain the phenotypes. Even if this set of data suggest that NUP153 RBD impact translation in a E sequence dependent manner in vitro, it remains very hard to comprehend how the rate of translation would be changed when the ribosomes translate the sequences encoding non-structural proteins as part of the same open reading frame. In line 64 and 475, the authors mention that the results “indicate NUP153 involvement in translation elongation” but this not addressed in this study per se. Ribosome profiling would be the gold standard if the authors want to make such a conclusion. So far, the data only indicated that site 1 in E impact translation in vitro when NUP153 RBD is added. It might also be more complex in cells, when the ER is present and remodelled by the virus. Without additional experiments addressing translation elongation, the conclusion is not supported by the data in my view. It is more likely that NUP153 impact the viral protein abundance at the posttranslational level, which was not addressed. The half-life the viral proteins could be tested by western blotting in wt and NUP153-depleted cells. This should not change according to the model elaborated by the authors.

Thank you for your insightful comments and suggestions, we have now removed translation “elongation” throughout the manuscript. As we cannot pinpoint that elongation is the target and precise mechanism of NUP153s proviral function. We have also analyzed the

half-life of the different viral proteins after cycloheximide treatment at different timepoints post treatment. Here we do not find a difference in stability of the viral proteins between wt and NUP153 depleted cells, this data is now new panels in Fig 8J and Supp fig 8. We have modified the results section to include the data line 394-397.

“Furthermore, to exclude that the effects we observed are a result of decreased protein stability following NUP153 depletion, we performed a cycloheximide (CHX) assay on TBEV infected cells and found no significant differences in the stability of the viral proteins E, NS1 and NS2B between auxin treated and untreated samples over time (Fig 8J and Supp Fig 8F,G).

Other comments:

-In Fig 4D, even if the data have been normalized to the input, these experiments tend to show that NUP153 associate with the viral negative strand (actually in a similar extent as the positive strand if the upper data point is not considered in the far right series). Even if the negative strand was not detected in the CLIP-seq data, the conclusion about the exclusive binding to the positive strand should be tuned down.

Thank you for your suggestion we have now adjusted the text to better reflect our results, line 223 and 445.

- The quality of Fig S7D is not optimal, and the results are hard to appreciate for the reader.

We agree that the fig S7D blot is a bit hard to interpret, but it is particularly difficult to detect in vitro translated proteins using WB, that is why we opted to use a luciferase system to have a more sensitive and quantitative method for our assay and include the WB as a supporting data piece. However, we do find that the provided WB reflects and supports the findings in the luciferase assay.

-Fig S8C is a bit confusing. What is the difference between the multiple “+” lanes? Are they biological replicates?

Thank you for pointing this out, the + lanes are TBEV infected samples and we show all the biological replicas in each blots, to be able to quantify and compare. We have now clarified it in the figure legend line 1449.

Reviewer #3 (Remarks to the Author):

The authors have addressed the majority of my concerns and added significant new experimentation and compelling evidence to support their claims. There are some outstanding questions/comments to address:

1. There is not a significant explanation as to why auxin addition after 4 hpi would have no impact. Why would this mechanism not be effective later in the infection cycle, closer to when C-prME are required for virion assembly?

Thank you for the comment. Auxin addition at 4 h post infection (NUP153 depletion at 6 h) do actually give an effect on infection Fig 2I-J, but upon addition of Auxin at 6 hpi (depletion at 8 h ; Fig 2 K and Supp Fig 2E) we no longer see an effect. To better understand the kinetics when NUP153 is required for virus infection we have performed additional experiments. We analyze the viral replication by Q-PCR assay at different timepoint post infection (2, 8, 12 and 16) in cells expressing NUP153 or AID cells depleted of NUP153. The RNA levels were normalized to the input timepoint 2 h, and we show that there is no effect on viral replication 8 and 12 h post infection (New Fig 2L). We believe that this is because NUP153 is important early in infection for the first round of viral particle release to promote viral propagation. We further validate this in Fig 8K with the BFA assay where we observe the delay in infection.

2. The data suggests a mechanism controlling translation; however, there are no data examining differences in protein stability. This caveat could be addressed in the text.

Thank you for this suggestion. To address protein stability, we have performed an additional experiment where we treated infected cells with cycloheximide to block protein translation, and analyzed protein amount in the cell using WB. Auxin was added 2 h prior infection and cycloheximide was added 24 h post infection. This is now new figure panels Fig 8J and Fig S8F-G, lines 393-398.

“ Furthermore, to exclude that the effects we observed was a result of decreased protein stability following NUP153 depletion, we performed a cycloheximide (CHX) assay on TBEV infected cells and found no significant differences in the stability of the viral proteins E, NS2B and NS3 between auxin treated and untreated samples over time (Fig 8J and Supp Fig 8F,G). However, we did observe a small change in NS1 expression 6 hours post CHX treatment (Supp Fig 8F,G). Taken together, .”

3. In Figure 4G, RNA binding may be somewhat dependent on the protein expression levels of the various truncated constructs. Expression of eGFP seems to be variable by IF in Figure

4B. Is RNA binding normalized to protein expression? A western blot showing relative protein expression of the constructs should be included.

Thank you for this suggestion. Assuming that the comment relates to figure 4C, where we do clip assay on different truncated constructs, this is a valid point. The RNA levels are not normalized to protein levels in the cell, however, we do not believe that the protein amount of the mutants play a major role in the results obtained with the clip assay. The reason being: When we analyze the protein expression levels of the different transfected constructs, they are somewhat different as indicated in the comments. However, the expression levels cannot explain the difference detected in the clip assay. NUP153wt, 1-650 and 1-650-RBD show similar protein levels; NUP153wt and 1-650 bind RNA, whereas 1-650-RBD does not bind viral RNA; similarly 880-1475 and 1-250 show comparable protein expression 880-1475 bind but 1-250 does not bind viral RNA.

We have added a western blot showing this as new panel A in supplementary fig 4.

Line 211 “First protein expression of the NUP153-GFP constructs were verified by WB (Supp Fig 4A). “

4. The paragraph explaining the binding peaks and the mutants generated is difficult to understand (lines 251-267). A more clear explanation would be helpful.

Thank you for this suggestion we have modified the text to be more clearly describe the experiments and results.

“The RNA-binding domain of NUP153 has previously been shown to associate with G- and GC-rich regions of RNA molecule^{12,42}. G-rich sequences can fold into G-quadruplex (G4) structures, a type of four-stranded nucleic acids secondary structure that can fold in certain G-rich sequences (43). Peak 1 contains two predicted G4 structures (Fig 5D boxes and Supp Fig 6A). The second of which is highly conserved among orthoflaviviruses sequences (Supp Fig 4E) (44,45). We used in vitro transcription to generate RNA fragments containing the wildtype (WT) peak 1, a mutated construct (11G1C) lacking the second highly conserved G4 motif and a construct lacking all orthoflavivirus conserved Gs (28G) (Fig 5D). We also in vitro transcribed three versions of peak 2, the WT peak, a mutated construct lacking the GA pattern that we identified to be conserved among orthoflaviviruses (8G8A) and a construct lacking all conserved Gs (19G) (Supp Fig 5A). RNA-EMSA was used to determine the interaction of the NUP153 RBD with the different RNA fragments corresponding to peak 1 and 2 and their mutants. We found that WT fragments of peaks 1 and 2 bound better to the NUP153 RBD compared to the 28G and 19G

fragments respectively that lacked all conserved Gs (Fig 5E-H and Supp Fig 5B-C), indicating that Gs in these regions are important for the interaction with NUP153.”

5. Figure 2G – while the interaction between NUP153 and Capsid is diminished with RNase treatment, it is not abolished as stated in the manuscript (Lines 179-181). Further, it appears that RNase treatment results in an overall decrease in capsid expression in the lysate.

Thank you for this comment. We have now changed the description of our results to better represent the data shown. The reason for the lower C levels during RNase A treatment in the input might be due to decreased stability of the C protein in the absence of RNA. We have modified the text accordingly line 182-186.

Line 178-183: “All three viral proteins co-immunoprecipitated with NUP153 in the presence of RNA, but the interaction between NUP153 and C was reduced after RNase A treatment, indicating that NS3 and NS5 bind to NUP153 independent of RNA whereas the C interaction is likely is indirect via RNA dependent. Furthermore, we also find that RNase A treatment resulted in decreased C protein in the input (Fig 3G), suggesting that RNA could be important for this protein stability.”

Response to Reviewers

Reviewer #1 (Remarks to the Author):

The additional data provided by the authors significantly enhance confidence in their conclusions, and the revisions to the manuscript satisfactorily address my major concerns.

Thank you so much.

Reviewer #2 (Remarks to the Author):

In this revised version of the manuscript, Peters and colleague have notably complemented the study with data showing stability of viral proteins over 6 hours and replication kinetics in wt and NUP153-deficient cells. Unfortunately, while these data are informative, they do not contribute to firmly conclude that NUP153 regulates the translation of structural proteins to favor viral particle production, the main claim of the work. Although all aspects related to elongation have been removed from the text, a potential role of NUP153 on translation still only relies on in vitro assays and reporter constructs far from the reality of a flaviviral genome. Thus, my previous concern remains, especially since the new results from stability assays as such (Fig 8 and Suppl Fig 8) do not allow to measure the half-life of the viral proteins as they look quite stable over the 6 hour time frame. Again, direct evidence should support this claim such as for instance: 1- evaluating vRNA distribution change in the fraction of a polysome sucrose gradient, or 2- viral protein/reporter expression following the transfection of a replication-deficient full-length viral genome (which does not require high transfection efficiency).

Thank you for your suggestions we have now adjusted the text throughout to not overstate our claims regarding NUP153s ability to modulate protein translation.

Reviewer #3 (Remarks to the Author):

The authors have provided sufficient experimentation to address the majority of reviewer concerns. Additional studies, including replicon-based translation assays and more in-depth work to quantify viral protein half-life in the context of infection would further support a role for NUP153 in specifically regulating translation. However, this is an informative study that convincingly shows that NUP153 is a pro-viral factor in flavivirus infection, which impacts the ratio of structural to nonstructural proteins and posits an intriguing model for translational control in flavivirus infection.

Thank you for your kind words.